# Robust Learning for Smoothed Online Convex Optimization with Feedback Delay

**Pengfei Li**
University of California Riverside
Riverside, CA, USA
pli081@ucr.edu

**Jianyi Yang**
University of California Riverside
Riverside, CA, USA
jyang239@ucr.edu

**Adam Wierman**
California Institute of Technology
Pasadena, CA, USA
adamw@caltech.edu

**Shaolei Ren**
University of California Riverside
Riverside, CA, USA
shaolei@ucr.edu

## Abstract

We study a challenging form of Smoothed Online Convex Optimization, a.k.a. SOCO, including multi-step nonlinear switching costs and feedback delay. We propose a novel machine learning (ML) augmented online algorithm, Robustness-Constrained Learning (RCL), which combines untrusted ML predictions with a trusted expert online algorithm via constrained projection to robustify the ML prediction. Specifically, we prove that RCL is able to guarantee $(1+\lambda)$-competitiveness against any given expert for any $\lambda > 0$, while also explicitly training the ML model in a robustification-aware manner to improve the average-case performance. Importantly, RCL is the first ML-augmented algorithm with a provable robustness guarantee in the case of multi-step switching cost and feedback delay. We demonstrate the improvement of RCL in both robustness and average performance using battery management for electrifying transportation as a case study.

## 1 Introduction

This paper studies *Smoothed Online Convex Optimization (SOCO)*, a model that has seen application in a wide variety of settings. The goal of SOCO is to minimize the sum of a per-round hitting cost and a switching cost that penalizes temporal changes in actions. The added (even single-step) switching cost creates substantial algorithmic challenges, and has received more than a decade of attention (see [1, 2, 3, 4, 5, 6, 7, 8, 9, 10, 11, 12] and the references therein). While there have been various competitive online algorithms, e.g., ROBD, to guarantee the worst-case performance robustness for SOCO [13, 5, 2, 6, 8, 13], their average performance is typically far from optimal due to the conservativeness needed to address potentially adversarial instances. In contrast, machine learning (ML) based optimizers can improve the average performance by exploiting rich historical data and statistical information [14, 1, 12, 7, 15], but they sacrifice the strong robustness in terms of provable competitive bounds needed by safety-critical applications, especially when there is a distributional shift [16, 17], the ML model capacity is limited, and/or inputs are adversarial [18, 19].

More recently, *ML-augmented online algorithms* have emerged as potential game changers in classic online problems such as ski rental and caching systems [20, 21, 22, 23, 24]. The goal is to obtain the best of both worlds by utilizing good ML predictions to improve the average performance while ensuring bounded competitive ratios even when ML predictions are arbitrarily bad. In the context of SOCO, there has been initial progress on ML-augmented algorithms in the past year [1, 14, 7, 25]. However, these studies target the simplest case of SOCO where there is no feedback delay and the

37th Conference on Neural Information Processing Systems (NeurIPS 2023).

switching costs are linear. Crucially, their specific designs make it difficult, if not impossible, to apply to more general and practical settings where there is hitting cost feedback delay and multi-step nonlinear memory in the switching cost. In addition, with a few exceptions [12, 25], a common assumption in the existing SOCO studies is that the ML model is pre-trained as a black box without awareness of the downstream operation, which creates a mismatch between training and testing and degrades the average performance.

Even without ML predictions, addressing the hitting cost feedback delay and multi-step nonlinear memory is already challenging, as the agent must make decisions semi-*blindly* without receiving the immediate hitting cost feedback and the decision at each step can affect multiple future decisions in a complex manner due to multi-step nonlinear switching costs [6, 26]. Incorporating ML predictions into the decision process adds substantial challenges, requiring novel algorithmic techniques beyond those used in the simple SOCO setting with single-step memory and no feedback delay [1, 14, 7].

**Contributions.** We propose a novel ML-augmented algorithm, called Robustness-Constrained Learning (RCL) that, for the first time, provides both robustness guarantees and good average performance for SOCO in general settings with hitting cost feedback delay and multi-step nonlinear memory in the switching cost. The foundation of RCL is to utilize an existing online algorithm (referred to as expert) as well as a novel reservation cost to hedge against future risks while closely following the ML predictions. Without having the immediate hitting cost feedback, RCL robustifies untrusted ML predictions at each step by judiciously accounting for the hitting cost uncertainties and the non-linear impact of the current decision on future switching costs. Importantly, by design, the resulting cost of RCL is no greater than $(1 + \lambda)$ times the expert's cost for any $\lambda > 0$ and any problem instance, while a larger $\lambda$ allows RCL to better explore the potential of good ML predictions.

Our main technical results provide bounds on both the worst-case and average-case performance of RCL. In particular, we prove a novel worst-case cost bound on RCL in Theorem 4.1 and a bound on the average-case performance in Theorem 4.2. Our cost bound is proven by utilizing a new reservation cost as the core of RCL. The form of the reservation cost allows us to develop a new proof approach (potentially of independent interest) that decouples the dependency of the online action on the history and constructs a new sufficient robustness constraint to bound the distance between the actions and ML predictions. Importantly, this approach enables us to account for multi-step non-linear memory in the switching cost and arbitrarily delayed hitting cost feedback in SOCO, which cannot be addressed by the existing algorithms [1, 14, 7, 12, 25]. We also provide a first-of-its-kind condition for simultaneously achieving both finite robustness and 1-consistency, which has been shown to be impossible in general [14]. Finally, we evaluate the performance of RCL using a case study focused on battery management in electric vehicle (EV) charging stations. Our results highlight the advantage of RCL in terms of robustness guarantees compared to pure ML-based methods, as well as the benefit of training a robustification-aware ML model.

In summary, our work makes significant contributions to the growing SOCO literature. First, we propose a novel ML-augmented algorithm that provides the first worst-case cost bounds in a general SOCO setting with hitting cost feedback delay and multi-step non-linear switching costs. Our algorithm design and analysis (Theorem 4.1) are new and significantly differ from those used in simple SOCO settings [1, 14, 7, 12, 25]. Second, we provide a first sufficient condition under which finite robustness and 1-consistency are simultaneously achieved (Corollary 4.1.1). Finally, we introduce and analyze the first algorithm that allows robustification-aware training, highlighting its advantage over the commonly-assumed robustification-oblivious training in terms of the average cost.

## 2   Related Work

SOCO has been actively studied for over a decade under a wide variety of settings [2, 3, 4, 5, 6, 7, 8, 9, 10, 11, 12]. For example, designed based on classic algorithmic frameworks, expert online algorithms include online gradient descent (OGD) [27], online balanced descent (OBD) [9], regularized OBD (ROBD) [13, 5], among many others. These algorithms are judiciously designed to have bounded competitive ratios and/or regrets, but they may not perform well on typical instances due to the conservative choices necessary to optimize the worst-case performance. Assuming the knowledge of (possibly imperfect) future inputs, algorithms include standard receding horizon control (RHC) [10] committed horizon control (CHC) [28], and receding horizon gradient descent (RHGD) [29, 4]. Nonetheless, the worst-case performance is still unbounded when the inputs have large errors.

By tapping into historical data, pure ML-based online optimizers, e.g., recurrent neural networks, have been studied for online problems [30, 31, 32]. Nonetheless, even with (heuristic) techniques such as distributionally robust training and/or addition of hard training instances (e.g., adversarial samples) [17, 16], they cannot provide formal worst-case guarantees as their expert counterparts.

By combining potentially untrusted ML predictions with robust experts, ML-augmented algorithms have emerged as a promising approach [22, 24, 33, 20]. The existing ML-augmented algorithms for SOCO [25, 1, 14, 7, 15, 12] only focus on simple SOCO settings where the hitting cost is known without delays and the switching cost is linear. Extending these algorithms [25, 1, 14, 7, 15, 12] to the general SOCO setting requires substantially new designs and analysis. For example, [25] utilizes the simple triangle inequality for linear switching costs in the metric space to achieve robustness, whereas this inequality does not hold given (multi-step) non-linear memory in terms of squared switching costs [14] even when there is no feedback delay. In fact, even without considering ML predictions, the general SOCO setting with feedback delays and multi-step non-linear switching costs presents significant challenges that need new techniques [26, 6] beyond those for the simple SOCO setting. Thus, RCL makes novel contributions to the growing ML-augmented SOCO literature. Customizing ML to better suit the downstream operation to achieve a lower cost has been considered for a few online problems [34, 35, 25]. In RCL, however, we need implicit differentiation through time to customize ML by considering our novel algorithm designs in our general SOCO setting.

In online learning with expert predictions [36, 37, 38], experts are dynamically chosen with time-varying probabilities to achieve a low regret compared to the best expert in hindsight. By contrast, RCL considers a different problem setting with feedback delay and multi-step non-linear memory, and focuses on constrained learning by bounding the total cost below $(1 + \lambda)$ times of the expert's cost for any instance and any $\lambda > 0$. Finally, RCL is also broadly relevant to conservative bandits and reinforcement learning [39]. Specifically, conservative exploration focuses on unknown cost functions (and, when applicable, transition models) and uses a baseline policy to guide the exploration process. But, its design is fundamentally different in that it does not hedge against future uncertainties when choosing an action for each step. Additionally, constrained policy optimization [40, 41] focuses on constraining the *average* cost, whereas RCL focuses on the worst-case cost constraint.

## 3 Model and Preliminaries

In a SOCO problem, an agent, a.k.a., decision maker, must select an irrevocable action $x_t$ from an action space $\mathcal{X} \subseteq \mathbb{R}^n$ with size $|\mathcal{X}|$ at each of time $t = 1, \ldots, T$. Given the selected action, the agent incurs the sum of (i) a non-negative hitting cost $f(x_t, y_t) \geq 0$ parameterized by the context $y_t \in \mathcal{Y} \subseteq \mathbb{R}^m$, where $f(\cdot) : \mathbb{R}^n \to \mathbb{R}_{\geq 0}$, and (ii) a non-negative switching cost $d(x_t, x_{t-p:t-1}) = \frac{1}{2}\|x_t - \delta(x_{t-p:t-1})\|^2$, where the constant $\frac{1}{2}$ is added for the convenience of derivation, $\|\cdot\|$ is the $l_2$ norm by default, and $\delta(\cdot) : \mathbb{R}^{p \times n} \to \mathbb{R}^n$ is a (possibly non-linear) function of $x_{t-p:t-1} = (x_{t-p}, \cdots, x_{t-1})$. We make the following standard assumptions.

**Assumption 1.** *At each $t$, the hitting cost $f(x_t, y_t)$ is non-negative, $\alpha_h$-strongly convex, and $\beta_h$-smooth in $x_t \in \mathcal{X}$. It is also Lipschitz continuous with respect to $y_t \in \mathcal{Y}$.*

**Assumption 2.** *In the switching cost $d(x_t, x_{t-p:t-1}) = \frac{1}{2}\|x_t - \delta(x_{t-p:t-1})\|^2$, the function $\delta(x_{t-p:t-1})$ is $L_i$-Lipschitz continuous in $x_{t-i}$ for $i = 1 \cdots p$, i.e., for any $x_{t-i}, x'_{t-i} \in \mathcal{X}$, we have $\|\delta(x_{t-p}, \cdots, x_{t-i}, \cdots x_{t-1}) - \delta(x_{t-p}, \cdots, x'_{t-i}, \cdots x_{t-1})\| \leq L_i\|x_{t-i} - x'_{t-i}\|$.*

The convexity of the hitting cost is standard in the literature and needed for competitive analysis, while smoothness (i.e., Lipschitz-continuous gradients) guarantees that bounded action differences also result in bounded cost differences [26]. A common example of the hitting cost is $f(x_t, y_t) = \|x_t - y_t\|^2$ as motivated by object tracking applications, where $y_t$ is the online moving target [6, 26]. In the switching cost term, the previous $p$-step actions $x_{t-p:t-1}$ are essentially encoded by $\delta(x_{t-p:t-1})$ [6]. Let us take drone tracking as an example. The switching cost can be written as $d(x_t, x_{t-1}) = \frac{1}{2}\|x_t - x_{t-1} + C_1 + C_2 \cdot |x_{t-1}| \cdot x_{t-1}\|^2$ and hence $\delta(x_{t-1}) = x_{t-1} - C_1 - C_2 \cdot |x_{t-1}| \cdot x_{t-1}$ is nonlinear, where $x_t$ is the drone's speed at time $t$ and the constants of $C_1$ and $C_2$ account for gravity and the aerodynamic drag [42]. For additional examples of switching costs in other applications, readers are further referred to [26].

For the convenience of presentation, we use $y = (y_1, \cdots, y_T) \in \mathcal{Y}^T$ to denote a problem instance, while noting that the initial actions $x_{-p+1:0} = (x_{-p+1}, \cdots, x_0)$ are also provided as an additional

input. Further, for $1 \leq t_1 \leq t_2 \leq T$, we also rewrite $\sum_{\tau=t_1}^{t_2} f(x_\tau, y_\tau) + d(x_\tau, x_{\tau-p:\tau-1})$ as $\text{cost}(x_{t_1:t_2})$, where we suppress the context $y_t$ without ambiguity.

For online optimization, the key challenge is that the switching cost couples online actions and the hitting costs are revealed online. As in the recent literature on SOCO [26], we assume that the agent knows the switching cost, because it is determined endogenously by the problem itself and the agent's previous actions. The agent also knows the smoothness constant $\beta_h$, although the hitting cost function itself is revealed online subject to a delay as defined below.

### 3.1 Feedback Delay

There may be feedback delay that prevents immediate observation of the context $y_t$ (which is equivalent to delayed hitting cost function) [43, 6]. For example, in assisted drone landing, the context parameter $y_t$ can represent the target velocity at time $t$ sent to the drone by a control center, but the communications between the drone and control center can experience delays due to wireless channels and/or even packet losses due to adversarial jamming [26, 42].

To model the delay, we refer to $q \geq 0$ as the maximum feedback delay (i.e., context $y_t$ can be delayed for up to $q \geq 0$ steps), and define $q$-delayed time set of arrival contexts.

**Definition 1** ($q$-delayed time set of arrival contexts). *Given the maximum feedback delay of $q \geq 0$, for each time $t = 1, \ldots, T$, the q-delayed time set of arrival contexts contains the time indexes whose contexts are newly revealed to the agent at time $t$ and is defined as $\mathcal{D}_t^q \subseteq \{\tau \in \mathbb{N} \mid t - q \leq \tau \leq t\}$ such that $\{\tau \in \mathbb{N} \mid \tau \leq t - q\} \subseteq (\mathcal{D}_1^q \bigcup \cdots \bigcup \mathcal{D}_t^q)$.*

Naturally, given the maximum delay of $q \geq 0$, we must have $\{\tau \in \mathbb{N} \mid \tau \leq t - q\} \subseteq (\mathcal{D}_1^q \bigcup \cdots \bigcup \mathcal{D}_t^q)$, i.e., at time $t$, the agent must have already known the contexts $y_\tau$ for any $\tau = 1, \cdots, t - q$.

It is worth highlighting that our definition of $\mathcal{D}_t^q$ is flexible and applies to various delay models. Specifically, the no-delay setting corresponds to $q = 0$ and $D_t^{q=0} = \{t\}$, while $q = T$ captures the case in which the agent may potentially have to choose actions without knowing any of the contexts $y_1, \cdots, y_T$ throughout an entire problem instance. Given the maximum delay $q \geq 0$, the delayed contexts can be revealed to the agent in various orders different from their actual time steps, i.e., the agent may receive $y_{t_1}$ earlier than $y_{t_2}$ for $t_1 > t_2$. Also, the agent can receive a batch of contexts $y_{t-q}, \cdots, y_t$ simultaneously at time $t$, and receive no new contexts some other time steps.

In online optimization, handling delayed cost functions, even for a single step, is challenging [6, 44, 26]. Adding ML predictions into online optimization creates further algorithmic difficulties.

### 3.2 Performance Metrics

Our goal is to minimize the sum of the total hitting costs and switching costs over $T$ time steps: $\min_{x_1, \cdots x_T} \sum_{t=1}^T f(x_t, y_t) + d(x_t, x_{t-p:t-1})$. We formally define our two measures of interest.

**Definition 2** (Competitiveness). *An algorithm $ALG_1$ is said to be $CR$-competitive against another baseline algorithm $ALG_2$ if $cost(ALG_1, y) \leq CR \cdot cost(ALG_2, y)$ is satisfied for any problem instance $y \in \mathcal{Y}^T$, where $cost(ALG_1, y)$ and $cost(ALG_2, y)$ denote the total costs of $ALG_1$ and $ALG_2$, respectively.*

**Definition 3** (Average cost). *The average cost of an algorithm $ALG$ is $\overline{cost}(ALG) = \mathbb{E}_{y \in \mathbb{P}_y}[cost(ALG, y)]$, where $cost(ALG, y)$ denotes the cost of $ALG$ for a problem instance $y$, and $\mathbb{P}_y$ is the exogenous probability distribution of $y = (y_1, \cdots, y_T) \in \mathcal{Y}^T$.*

Our definition of competitiveness against a general baseline algorithm is commonly considered in the literature, e.g., [7]. The two metrics measure an online algorithm's robustness in the worst case and expected performance in the average case, which are both important in practice.

We consider an expert (online) algorithm $\pi$ which chooses $x_t^\pi$ at time $t$ and an ML model $h_W$ which, parameterized by $W$, produces $\tilde{x}_t = h_W(\tilde{x}_{t-p:t-1}, \{y_\tau \mid \tau \in \mathcal{D}_t^q\})$ at time $t$. As in the existing ML-augmented online algorithms [7, 14], RCL chooses an actual online action $x_t$ by using the two actions $x_t^\pi$ and $\tilde{x}_t$ as advice. In general, it is extremely challenging, if not impossible, to simultaneously optimize for both the average cost and the competitiveness. Here, given a robustness requirement $\lambda > 0$, we focus on minimizing the average cost while ensuring $(1 + \lambda)$-competitiveness against

the expert $\pi$. Crucially, the optimal expert for our setting is iROBD, which has the best-known competitiveness against the offline optimal $OPT$ with complete information [26]. Thus, by using iROBD as the expert $\pi$, $(1+\lambda)$-competitiveness of RCL against $\pi$ immediately translates into a scaled competitive ratio of $(1+\lambda) \cdot CR_{iROBD}$ against $OPT$, where $CR_{iROBD}$ is iROBD's competitive ratio against $OPT$.

## 4  RCL: The Design and Analysis

In this section, we present RCL, a novel ML-augmented that combines ML predictions (i.e., online actions produced by an ML model [1, 45, 12, 15]) with a robust expert online algorithm to the worst-case cost while leveraging the benefit of ML predictions for average performance.

### 4.1  Robustness-Constrained Online Algorithm

Our goal is to "robustify" ML predictions, by which we mean that we want to ensure a robustness bound on the cost of no greater than $(1+\lambda)$ times of the expert's cost, i.e., for any problem instance $y$, we have $\mathrm{cost}(x_{1:T}) \leq (1+\lambda)\mathrm{cost}(x^\pi_{1:T})$, where $\lambda > 0$ is a hyperparameter indicating the level of robustness we would like to achieve. Meanwhile, we would like to utilize the benefits of ML predictions to improve the average performance.

Because of the potential feedback delays, RCL needs to choose an online action $x_t$ without necessarily knowing the hitting costs of the expert's action $x^\pi_t$ and ML prediction $\tilde{x}_t$. Additionally, the action $x_t$ can affect multiple future switching costs due to multi-step non-linear memory. Thus, it is very challenging to robustify ML predictions for the SOCO settings we consider. A simple approach one might consider is to constrain $x_t$ such that the cumulative cost up to each time $t$ is always no greater than $(1+\lambda)$ times of the expert's cumulative cost, i.e., $\mathrm{cost}(x_{1:t}) \leq (1+\lambda)\mathrm{cost}(x^\pi_{1:t})$. However, even without feedback delays, such an approach may not even produce feasible actions for some $t = 1, \cdots, T$. We explain this by considering a single-step switching cost case. Suppose that $\mathrm{cost}(x_{1:t}) \leq (1+\lambda)\mathrm{cost}(x^\pi_{1:t})$ is satisfied at a time $t < T$, and we choose an action $x_t \neq x^\pi_t$ different from the expert. Then, at time $t+1$, let us consider a case in which the expert algorithm has such a low cost that even choosing $x_{t+1} = x^\pi_{t+1}$ will result in $\mathrm{cost}(x_{1:t}) + f(x^\pi_{t+1}, y_{t+1}) + d(x^\pi_{t+1}, x_t) > (1+\lambda)\left[\mathrm{cost}(x^\pi_{1:t}) + f(x^\pi_{t+1}, y_{t+1}) + d(x^\pi_{t+1}, x^\pi_t)\right]$. This is because the actual switching cost $d(x^\pi_{t+1}, x_t)$ can be significantly higher than the expert's switching cost $d(x^\pi_{t+1}, x^\pi_t)$ due to $x_t \neq x^\pi_t$. As a result, at time $t+1$, it is possible that there exist no feasible actions that satisfy $\mathrm{cost}(x_{1:t+1}) \leq (1+\lambda)\mathrm{cost}(x^\pi_{1:t+1})$. Moreover, when choosing an online action close to the ML prediction, extra caution must be exercised as the hitting costs can be revealed with delays of up to $q$ steps.

To address these challenges, RCL introduces novel reservation costs to hedge against any possible uncertainties due to hitting cost feedback delays and multi-step non-linear memory in the switching costs. Concretely, given both the expert action $x^\pi_t$ and ML prediction $\tilde{x}_t$ at time $t$, we choose $x_t = \arg\min_{x \in \mathcal{X}_t} \frac{1}{2}\|x - \tilde{x}_t\|^2$ by solving a constrained convex problem to project the ML prediction $\tilde{x}_t$ into a robustified action set $x \in \mathcal{X}_t$ where $x_t$ satisfies:

$$
\sum_{\tau \in \mathcal{A}_t} f(x_\tau, y_\tau) + \sum_{\tau=1}^{t} d(x_\tau, x_{\tau-p:\tau-1}) + \sum_{\tau \in \mathcal{B}_t} H(x_\tau, x^\pi_\tau) + G(x_t, x_{t-p:t-1}, x^\pi_{t-p:t})
$$
$$
\leq (1+\lambda)\left( \sum_{\tau \in \mathcal{A}_t} f(x^\pi_\tau, y_\tau) + \sum_{\tau=1}^{t} d(x^\pi_\tau, x^\pi_{\tau-p:\tau-1}) \right),
\tag{1}
$$

in which $\lambda > 0$ is the robustness hyperparameter, $\mathcal{A}^q_t = (\mathcal{D}^q_1 \bigcup \cdots \bigcup \mathcal{D}^q_t)$ and $\mathcal{B}^q_t = \{1, \cdots, t\} \backslash \mathcal{A}_t$ are the sets of time indexes for which the agent knows and does not know the context parameters up to time $t$, respectively. Most importantly, the two novel reservation costs $H(x_\tau, x^\pi_\tau)$ and $G(x_t, x_{t-q:t-1}, x^\pi_{t-q:t})$ are defined as

$$
H(x_\tau, x^\pi_\tau) = \frac{\beta_h}{2}(1 + \frac{1}{\lambda_0})\|x_\tau - x^\pi_\tau\|^2,
\tag{2}
$$
$$
G(x_t, x_{t-p:t-1}, x^\pi_{t-p:t}) = \frac{(1 + \frac{1}{\lambda_0})(1 + \sum_{k=1}^{p} L_k)}{2} \sum_{k=1}^{p}\left( L_k\|x_t - x^\pi_t\|^2 + \sum_{i=1}^{p-k} L_{k+i}\|x_{t-i} - x^\pi_{t-i}\|^2 \right),
\tag{3}
$$

---
**Algorithm 1** Online Optimization with RCL
---
**Require:** $\lambda > 0$, $\lambda_0 \in (0, \lambda)$, initial actions $x_{1-p:0}$, expert algorithm $\pi$, and ML model $h_W$
  1: **for** $t = 1, \cdots, T$
  2:     Receive a set of contexts $\{y_\tau | \tau \in \mathcal{D}_t^q\}$
  3:     Get the expert's action $x_t^\pi$ given its own history
  4:     Get $\tilde{x}_t = h_W(\tilde{x}_{t-p:t-1}, \{y_\tau | \tau \in \mathcal{D}_t^q\})$
  5:     Choose $x_t = \arg\min_{x \in \mathcal{X}_t} \frac{1}{2}\|x - \tilde{x}_t\|^2$ subject to the constraint (1)   //Robustification
---

where $\beta_h$ is the smoothness constant of the hitting cost in Assumption 1, $L_k$ is the Lipschitz constant of $\delta(x_{t-p:t-1})$ in Assumption (2), and $\lambda_0 \in (0, \lambda)$ with the optimum being $\lambda_0 = \sqrt{1+\lambda} - 1$ (Theorem 4.1). The computational complexity for projection into (1) is tolerable due to convexity.

The interpretation of $H(x_\tau, x_\tau^\pi)$ and $G(x, x_{t-q:t-1}, x_{t-q:t}^\pi)$ is as follows. If RCL's action $x_\tau$ deviates from the expert's action $x_\tau^\pi$ at time $\tau$ and the hitting cost is not known yet due to delayed $y_\tau$, then it is possible that RCL actually experiences a high but unknown hitting cost. In this case, to guarantee the worst-case robustness, we include an upper bound of the cost difference as the reservation cost such that $f(x_\tau, y_\tau) - (1 + \lambda)f(x_\tau^\pi, y_\tau) \leq H(x_\tau, x_\tau^\pi)$ regardless of the delayed $y_\tau \in \mathcal{Y}$. If $y_\tau$ has been revealed at time $t$ (i.e., $\tau \in \mathcal{A}_t$), then we use the actual costs instead of the reservation cost. Likewise, by considering the expert's future actions as a feasible backup plan in the worst case, the reservation cost $G(x, x_{t-p:t-1}, x_{t-p:t}^\pi)$ upper bounds the maximum possible difference in the future switching costs (up to future $p$ steps) due to deviating from the expert's action at time $t$.

With $H(x_\tau, x_\tau^\pi)$ and $G(x, x_{t-q:t-1}, x_{t-q:t}^\pi)$ as reservation costs in (1), RCL achieves robustness by ensuring that following the expert's actions in the future is always feasible, regardless of the delayed $y_t$. The online algorithm is described in Algorithm 1, where both the expert and ML model have the same online information $\{y_\tau | \tau \in \mathcal{D}_t^q\}$ at time $t$ and produce their own actions as advice to RCL.

## 4.2 Analysis

We now present our main results on the cost bound of RCL, showing that RCL can indeed maintain the desired $(1 + \lambda)$-competitiveness against the expert $\pi$ while exploiting the potential of ML predictions.

**Theorem 4.1** (Cost bound). *Consider a memory length $p \geq 1$ and the maximum feedback delay of $q \geq 0$. Given a context sequence $y = (y_1, \cdots, y_T)$, let $cost(\tilde{x}_{1:T})$ and $cost(x_{1:T}^\pi)$ be the costs of pure ML predictions $\tilde{x}_{1:T}$ and expert actions $x_{1:T}^\pi$, respectively. For any $\lambda > 0$, by optimally setting $\lambda_0 = \sqrt{1+\lambda} - 1$, the cost of RCL is upper bounded by*

$$cost(x_{1:T}) \leq \min\left( (1 + \lambda)cost(x_{1:T}^\pi), \left( \sqrt{cost(\tilde{x}_{1:T})} + \sqrt{\frac{\beta_h + \alpha^2}{2}\Delta(\lambda)} \right)^2 \right), \qquad (4)$$

*where $\Delta(\lambda) = \sum_{i=1}^T \left[ \|\tilde{x}_t - x_t^\pi\|^2 - \frac{2(\sqrt{1+\lambda}-1)^2}{(\beta_h + \alpha^2)}cost_t^\pi \right]^+$ in which $cost_t^\pi = \left( \sum_{\tau \in \mathcal{D}_t^q} f(x_\tau^\pi, y_\tau) \right) + d(x_t^\pi, x_{t-p:t-1}^\pi)$ is the total of revealed hitting costs and switching cost for the expert at time $t$, $\beta_h$ is the smoothness constant of the hitting cost (Assumption 1), and $\alpha = 1 + \sum_{i=1}^p L_i$ with $L_1 \ldots L_p$ being the Lipschitz constants in the switching cost (Assumption 2).* ☐

Theorem 4.1 is the *first* worst-case cost analysis for ML-augmented algorithms in a general SOCO setting with delayed hitting costs and multi-step switching costs. Its proof is available in the appendix and outlined here. We prove the competitiveness against the expert $\pi$ based on our novel reservation cost $H(x_\tau, x_\tau^\pi)$ and $G(x_t, x_{t-q:t-1}, x_{t-q:t}^\pi)$ by induction. It is more challenging to prove the competitiveness against the ML prediction, because $x_t$ implicitly depends on all the previous actions of ML predictions and expert actions up to time $t$. To address this challenge, we utilize a novel technique by first removing the dependency of $x_t$ on the history. Then, we construct a new sufficient robustness constraint that allows an explicit expression of another robustified action, whose distance to the ML prediction $\tilde{x}_t$ is an upper bound by the distance between $x_t$ and $\tilde{x}_t$ due to the projection of $\tilde{x}_t$ into (1). Finally, due to the smoothness of the hitting cost function and the switching cost, the distance bound translates into the competitiveness of RCL against ML predictions.

The two terms inside the min operator in Theorem 4.1 show the tradeoff between achieving better competitiveness and more closely following ML predictions. To interpret this, we note that the first

term inside $\min$ operator shows $(1+\lambda)$-competitiveness of RCL against the expert $\pi$, while the second term inside $\min$ operator shows RCL can also exploit the potential of good ML predictions. A smaller $\lambda > 0$ means that we want to be closer to the expert for better competitiveness, while a larger $\lambda > 0$ decreases the term $\Delta(\lambda) = \sum_{i=1}^{T}\left[\|\tilde{x}_t - x_t^\pi\|^2 - \frac{2(\sqrt{1+\lambda}-1)^2}{(\beta_h+\alpha^2)}\mathrm{cost}_t^\pi\right]^+$ and hence makes RCL follow the ML predictions more closely.

The term $\Delta(\lambda)$ in (4) essentially bounds the total squared distance between the actual online action and ML predictions. Intuitively, RCL should follow the ML predictions more aggressively when the expert does not perform well. This insight is also reflected in $\Delta(\lambda)$ in Theorem 4.1. Concretely, when the expert $\pi$'s total revealed $\mathrm{cost}_t^\pi$ is higher, $\Delta(\lambda)$ also becomes smaller, pushing RCL closer to ML. On the other hand, when the expert's cost is lower, RCL stays closer to the better-performing expert for guaranteed competitiveness.

When $\|\tilde{x}_t - x_t^\pi\|^2$ is larger (i.e., greater discrepancy between the expert's action and ML prediction), it is naturally more difficult to follow both the expert and ML prediction simultaneously. Thus, given a robustness requirement of $\lambda > 0$, we see from $\Delta(\lambda)$ that a larger $\|\tilde{x}_t - x_t^\pi\|^2$ also increases the second term in the $\min$ operator in Theorem 4.1, making it more difficult for RCL to exploit the potential of ML predictions. Moreover, deviating from the expert's action at one step can have impacts on the switching costs in future $p$ steps. Thus, the memory length $p$ creates some additional friction for RCL to achieve a low cost bound with respect to ML predictions: the greater $p$, the greater $\alpha = 1 + \sum_{i=1}^{p} L_i$, and hence the greater the second term in (4).

**Robustness and consistency.** It remains to show the worst-case competitiveness against the offline optimal algorithm $OPT$, which is typically performed for two extreme cases — when ML predictions are extremely bad and perfect — which are respectively referred to as *robustness* and *consistency* in the literature [1, 20, 22] and formally defined below.

**Definition 4** (Robustness and consistency). *The robustness of RCL is $CR(\infty)$ if RCL is $CR(\infty)$-competitive against $OPT$ when the ML's competitiveness against $OPT$ is arbitrarily large (denoted as $\tilde{CR} \to \infty$) ; and the consistency of RCL is $CR(1)$ if RCL is $CR(1)$-competitive against $OPT$ when the ML's competitiveness against $OPT$ is $\tilde{CR} = 1$.*

Robustness indicates the worst-case performance of RCL for any possible ML predictions, whereas consistency measures the capability of RCL to retain the performance of perfect ML predictions. In general, the tradeoff between robustness and consistency is unavoidable for online algorithms [20, 14, 46]. The state-of-the-art expert algorithm iROBD recently proposed in [26] has the best-known competitive ratio under the assumption of identical delays for each context (i.e., $\mathcal{D}_t^q = \{t - q\}$ — each context is delayed by $q$ steps). The identical-delay model essentially ignores any other contexts $y_\tau$ for $\tau \in \{\tau \in \mathbb{N} \,|\, t - q + 1 \leq \tau \leq t\}$. Thus, it is the worst case of a general $q$-step delay setting, whose competitive ratio is upper bounded by that of iROBD. Consequently, due to $(1 + \lambda)$-competitiveness against any expert $\pi$ in Theorem 4.1, we immediately obtain a finite robustness bound for RCL by considering iROBD as the expert.

Nonetheless, even for the simplest SOCO setting with no feedback delay and a switching cost of $d(x_t, x_{t-1}) = \frac{1}{2}\|x_t - x_{t-1}\|^2$, a recent study [14] has shown that it is *impossible* to simultaneously achieve 1-consistency and finite robustness. Consequently, in general SOCO settings, the finite robustness of RCL given any $\lambda > 0$ means the impossibility of achieving 1-consistency by following perfect ML predictions without further assumptions.

Despite this pessimistic result due to the fundamental challenge of SOCO, we find a sufficient condition that can overcome the impossibility, which is formalized as follows.

**Corollary 4.1.1** (1-consistency and finite robustness). *Consider iROBD as the exert $\pi$, whose competitive ratio against $OPT$ is denoted as $CR_{iROBD}$ [26]. If the expert's switching cost always satisfies $d(x_t^\pi, x_{t-p:t-1}^\pi) \geq \epsilon > 0$ for any time $t = 1, \cdots, T$, then by setting $\lambda \geq \frac{|\mathcal{X}|^2(\alpha^2+\beta_h)}{2\epsilon} + \sqrt{\frac{2|\mathcal{X}|^2(\alpha^2+\beta_h)}{\epsilon}} \sim \mathcal{O}(\frac{1}{\epsilon})$ and optimally using $\lambda_0 = \sqrt{1+\lambda} - 1$, RCL achieves $(1+\lambda) \cdot CR_{iROBD}$-robustness and 1-consistency simultaneously.* $\square$

Corollary 4.1.1 complements the impossibility result for SOCO [14] by providing the first condition under which finite robustness and 1-consistency are simultaneously achievable. The intuition is that if the expert has a strictly positive switching cost no less than $\epsilon > 0$ at each time, then its per-step

cost is also no less than $\epsilon$, which provides RCL with the flexibility to choose a different action than the expert's action $x_t^\pi$ due to the $(1+\lambda)$ cost slackness in the competitiveness requirement. Therefore, by choosing a sufficiently large but still finite $\lambda \sim \mathcal{O}(\frac{1}{\epsilon})$, we can show that $\Delta(\lambda) = 0$ in (4) in Theorem 4.1, which means RCL can completely follow the ML predictions. Without this condition, it is possible that the expert's cost is zero in the first few steps, and hence RCL must follow the expert's actions at the beginning to guarantee $(1+\lambda)$-competitiveness in case the expert continues to have a zero cost in the future — even though ML predictions are perfect and offline optimal, RCL cannot follow them at the beginning because of the online process and $(1+\lambda)$-competitiveness requirement.

Importantly, our sufficient condition is not unrealistic in practice. For example, in moving object-tracking applications, the condition $d(x_\tau^\pi, x_{\tau-p:\tau-1}^\pi) \geq \epsilon > 0$ is satisfied if the expert's action $\tilde{x}_t$ keeps changing to follow the moving object over time or alternatively, we ignore the dummy time steps with no movement.

### 4.3 ML Model Training in RCL

We present the training details and highlight the advantage of training the ML model in a robustification-aware manner to reduce the average cost.

#### 4.3.1 Architecture, loss, and dataset

Because of the recursive nature of SOCO and the strong representation power of neural networks, we use a recurrent neural network with each base model parameterized by $W \in \mathcal{W}$ (illustrated in Fig. 1 in the appendix). Such architectures are also common in ML-based optimizers for other online problems [17, 31]. With historical data available, we can construct a training dataset $\mathcal{S}$ that contains a finite number of problem instances. The dataset can also be enlarged using data augmentation techniques (e.g., adding adversarial samples) [16, 17].

**Robustification-oblivious.** The existing literature on ML-augmented algorithms [22, 1, 45, 7] has commonly assumed that the ML model $h_W$ is separately trained in a robustification-oblivious manner without being aware of the downstream algorithm used online. Concretely, the parameter $W$ of a robustification-oblivious ML model is optimized for the following loss

$$W^* = \arg\min_{W \in \mathcal{W}} \frac{1}{|\mathcal{S}|} \sum_{\mathcal{S}} cost(\tilde{x}_{1:T}), \tag{5}$$

where $\tilde{x}_t = h_W(\tilde{x}_{t-p:t-1}, \{y_\tau | \tau \in \mathcal{D}_t^q\})$ is the ML prediction at time $t$.

**Robustification-aware.** There is a mismatch between the actual objective $cost(x_{1:T})$ and the training objective $cost(\tilde{x}_{1:T})$ of a robustification-oblivious ML model. To reconcile this, we propose to train the ML model in a robustification-aware manner by explicitly considering the robustification step in Algorithm 1. For notational convenience, we denote the actual action as $x_t = \mathsf{Rob}_\lambda(h_W) = \mathsf{Rob}_\lambda(\tilde{x}_t)$, which emphasizes the projection of $\tilde{x}_t$ into the robust action set (2). Thus, the parameter $W$ of a robustification-aware ML model is optimized to minimize the following loss

$$\hat{W}^* = \arg\min_{W \in \mathcal{W}} \frac{1}{|\mathcal{S}|} \sum_{\mathcal{S}} cost(\mathsf{Rob}_\lambda(\tilde{x}_{1:T})), \tag{6}$$

which is different from (5) that only minimizes the cost of pre-robustification ML predictions.

#### 4.3.2 Average cost

We bound the average cost of RCL given an ML model $h_W$.

**Theorem 4.2** (Average cost). *Assume that the ML model is trained over a dataset $\mathcal{S}$ drawn from the training distribution $\mathbb{P}_y'$. With probability at least $1 - \delta, \delta \in (0, 1)$, the average cost $\mathbb{E}_y[\mathrm{cost}(x_{1:T})]$ of RCL over the testing distribution $y \sim \mathbb{P}_y$ is upper bounded by*

$$\overline{cost}(RCL) \leq \min\left\{(1+\lambda)\overline{cost}(\pi), \overline{cost}_{\mathcal{S}}(\mathsf{Rob}_\lambda(h_W)) + \mathcal{O}\left(\mathrm{Rad}_{\mathcal{S}}(\mathsf{Rob}_\lambda(\mathcal{W})) + \sqrt{\frac{\log(\frac{2}{\delta})}{|\mathcal{S}|}}\right)\right\},$$

*where $\overline{cost}_{\mathcal{S}}(\mathsf{Rob}_\lambda(h_W)) = \frac{1}{|\mathcal{S}|}\sum_{\mathcal{S}} cost(x_{1:T})$ is the empirical average cost of robustified ML predictions in $\mathcal{S}$, $\mathrm{Rad}_{\mathcal{S}}(\mathsf{Rob}_\lambda(\mathcal{W}))$ defined in Definition 5 in the appendix is the Rademacher*

*complexity with respect to the ML model space parameterized by $\mathcal{W}$ with robustification on the training dataset $\mathcal{S}$, the scaling coefficient inside $\mathcal{O}$ for $\mathrm{Rad}_\mathcal{S}(\mathrm{Rob}_\lambda(\mathcal{W}))$ is $\Gamma_x = \sqrt{T}|\mathcal{X}| \left[\beta_h + \frac{1}{2}(1 + \sum_{i=1}^p L_i)(1 + \sum_{i=1}^p L_i)\right]$ with $|\mathcal{X}|$ being the size of the action space $\mathcal{X}$ and $\beta_h$, $L_i$, and $p$ as the smoothness constant, Lipschitz constant of the nonlinear term in the switching cost, and the memory length as defined in Assumptions 1 and 2, and the coefficient for $\sqrt{\frac{\log(\frac{2}{\delta})}{|\mathcal{S}|}}$ is $3\bar{c}$ with $\bar{c}$ being the upper bound of the total cost for an episode.*

Theorem 4.2 bounds the average cost of RCL by the minimum of two bounds. The first bound $(1 + \lambda)\overline{cost}(\pi)$ further highlights the guaranteed $(1 + \lambda)$-competitiveness of RCL with respect to the expert's average cost $AVG(\pi)$. The second bound includes a term $\overline{cost}_\mathcal{S}(\mathrm{Rob}_\lambda(h_W)) = \frac{1}{|\mathcal{S}|} \sum_\mathcal{S} cost(x_{1:T})$, which is the empirical average cost of RCL given an ML model $h_W$ and decreases when $\lambda > 0$ increases. The reason is that with a larger $\lambda > 0$, the robust action set (1) is enlarged and RCL has more freedom to follow ML predictions due to the less stringent competitiveness constraint. Given an ML model $h_W$, Theorem 4.1 shows how the upper bound on $\overline{cost}_\mathcal{S}(\mathrm{Rob}_\lambda(h_W))$ varies with $\lambda$. Note that $h_{W^*}$ in (5) and $h_{\hat{W}^*}$ in (6) minimize $\overline{cost}_\mathcal{S}(h_{W^*})$ and $\overline{cost}_\mathcal{S}(\mathrm{Rob}_\lambda(h_{\hat{W}^*}))$, respectively, while the post-robustication cost (i.e., $\overline{cost}_\mathcal{S}(\mathrm{Rob}_\lambda(h_{\hat{W}^*}))$) is the actual cost of RCL. Thus, we can further reduce the average cost by using the optimal robustification-aware ML model $h_{\hat{W}^*}$, compared to a robustification-oblivious model $h_{W^*}$.

The other terms inside the second bound in (7) are related to the training dataset: the larger dataset, the smaller Rademacher complexity $\mathrm{Rad}_\mathcal{S}(\mathrm{Rob}(\mathcal{W}))$ and $\sqrt{\frac{\log(1/\delta)}{|\mathcal{S}|}}$. Note that the Rademacher complexity $\mathrm{Rad}_\mathcal{S}(\mathrm{Rob}(\mathcal{W}))$ of RCL is no greater than that of using the ML model alone (i.e. $\mathrm{Rad}_\mathcal{S}(\mathrm{Rob}(\mathcal{W})) \leq \mathrm{Rad}_\mathcal{S}(\mathcal{W})$, shown in the appendix). The intuition is that RCL limits the action space for robustness. Thus, Theorem 4.1 provides the insight that, given $\lambda > 0$, robustification in RCL is more valuable in terms of bounding the average cost when the ML model $h_W$ is not well trained (e.g., due to inadequate training data). In practice, the hyperparameter $\lambda > 0$ can be set to improve the empirical average performance subject to the robustness constraint based on a held-out validation dataset along with the tuning of other hyperparameters (e.g., learning rates).

### 4.3.3 Robustification-aware training and experimental verification

Despite the advantage in terms of the average cost, it is non-trivial to train a robustification-aware ML model using standard back-propagation. This is because the operator of projecting ML predictions into a robust set (1) is a recursive *implicit* layer that cannot be easily differentiated as typical neural network layers. Due to space limitations, we defer to the appendix the the differentiation of the loss function with respect to ML model weights $W$.

We validate the theoretical analysis by exploring the empirical performance of RCL using a case study of battery management in electric vehicle (EV) charging stations [47]. Our results are presented in the appendix. The results highlight the advantage of RCL in terms of robustness compared to pure ML models, as well as the benefit of using a robustification-aware ML model in terms of the average cost.

## 5   Experiments

We now explore the performance of RCL using a case study focused on battery management in electric vehicle (EV) charging stations [48]. We first formulate the problem as an instance of SOCO. More details can be found at Appendix B.1. Then, we test RCL on a public dataset provided by ElaadNL, compared with several baseline algorithms including ROBD [26], EC-L2O [12], and HitMin (details in Appendix B.2). Our results highlight the advantage of RCL in terms of robustness guarantees compared to pure ML models, as well as the benefit of training a robustification-aware ML model in terms of the average cost.

### 5.1   Results

We now present some results for the case in which the hitting cost function (parameterized by $y_t$) is immediately known without feedback delay. The results for the case with feedback delay are presented in Appendix B.4. Throughout the discussion, the reported values are normalized with

| | RCL | | | | RCL-O | | | | ML | EC-L2O | ROBD | HitMin |
|---|---|---|---|---|---|---|---|---|---|---|---|---|
| | $\lambda$=0.6 | $\lambda$=1 | $\lambda$=3 | $\lambda$=5 | $\lambda$=0.6 | $\lambda$=1 | $\lambda$=3 | $\lambda$=5 | | | | |
| AVG | 1.4704 | 1.1144 | 1.0531 | **1.0441** | 1.4780 | 1.2432 | 1.0855 | 1.0738 | 1.0668 | 1.1727 | 1.6048 | 1.2003 |
| CR | 1.7672 | **1.2905** | 1.4405 | 1.3014 | 2.2103 | 2.4209 | 2.4200 | 3.0322 | 3.2566 | 2.0614 | 1.7291 | 2.0865 |

Table 1: Competitive ratio and average cost comparison of different algorithms.

respect to those of the respective OPT. The average cost (**AVG**) and competitive ratio (**CR**) are all empirical results reported on the testing dataset.

By Theorem 4.1, there is a trade-off (governed by $\lambda > 0$) between exploiting ML predictions for good average performance and following the expert for robustness. Here, we focus on the default setting of $\lambda = 1$ and discuss the impact of different choices of $\lambda$ on both RCL and RCL-O in Appendix B.3.2. As shown in Table 1, with $\lambda = 1$, both RCL and RCL-O have a good average cost, but RCL has a lower average cost than RCL-O and is outperformed only by ML in terms of the average cost. RCL and RCL-O have the same competitive ratio (i.e., $(1 + \lambda)$ times the competitive ratio of ROBD). Empirically, RCL has the lowest competitive ratio among all the other algorithms, demonstrating the practical power of RCL for robustifying, potentially untrusted, ML predictions. In this experiment, RCL outperforms ROBD in terms of the empirical competitive ratio because it exploits the good ML predictions for those problem instances that are adversarial to ROBD. This result complements Theorem 4.1, where we show theoretically that RCL can outperform ROBD in terms of the average cost by properly setting $\lambda$. By comparison, ML performs well on average by exploiting the historical data, but has the highest competitive ratio due to its expected lack of robustness. The two alternative baselines, EC-L2O and HitMin, are empirically good on average and also in the worst case, but they do not have guaranteed robustness. On the other hand, ROBD is very robust, but its average cost is also the worst among all the algorithms under consideration.

More results, including using HitMin as the expert and large distributional shifts, are available in Appendix B.3.

# 6 Conclusion

We have considered a general SOCO setting (including multi-step switching costs and delayed hitting costs) and proposed RCL, which ensures a worst-case performance bound by utilizing an expert algorithm to robustify untrusted ML predictions. We prove that RCL is able to guarantee $(1 + \lambda)$-competitive against any given expert for any $\lambda > 0$. Additionally, we provide a sufficient condition that achieves finite robustness and 1-consistency simultaneously. To improve the average performance, we explicitly train the ML model in a robustification-aware manner by differentiating the robustification step, and provide an explicit average cost bound. Finally, we evaluate RCL using a case study of battery management for EV stations, which highlights the improvement in both robustness and average performance compared to existing algorithms.

**Limitations and future work.** We discuss two limitations of our work. First, we make a few assumptions (e.g., convexity and smoothness) on the hitting costs and switching costs. While they are common in the SOCO literature, these assumptions may not always hold in practice, and relaxing them will be interesting and challenging. Second, for online optimization, we only consider the ML prediction for the current step, whereas predictions for future steps can also be available in practice and may be explored for performance improvement. There are also a number of interesting open questions that follow from this work. For example, it is interesting to study alternative reservation costs and the optimality in terms of the tradeoff between bi-competitiveness. Additionally, it would also be interesting to extend RCL to other related problems such as convex body chasing and metrical task systems.

**Acknowledgement.** We would like to thank the anonymous reviewers for their helpful comments. Pengfei Li, Jianyi Yang and Shaolei Ren were supported in part by the U.S. NSF under the grant CNS–1910208. Adam Wierman was supported in part by the U.S. NSF under grants CNS-2146814, CPS-2136197, CNS-2106403, NGSDI-2105648.

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

# A  Illustration of RCL

We illustrate the online optimization process of RCL in Fig. 1.

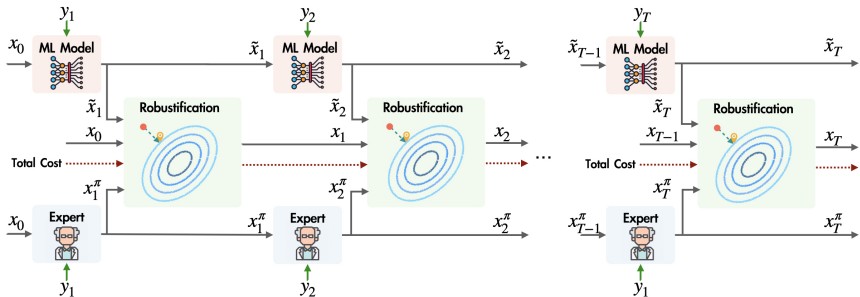

Figure 1: Robustness-constrained online optimization using RCL. The expert algorithm and ML model run independently. At each time $t = 1, \cdots, T$, RCL projects the ML prediction $\tilde{x}_t$ into a robustified action set.

# B  Case Study: Battery Management for EV Charging Stations

## B.1  Problem Formulation

Batteries are routinely used in EV charging stations to handle the rapidly fluctuating charging demands and protect the connected grid. Thus, properly managing battery charging/discharging decisions is crucial for reliability, lifespan, and safety of batteries and grids.

We consider the management of $N$ batteries. At each time step $t$, suppose that $x_t \in \mathbb{R}_+^N$ represents the State of Charge (SoC) and $u_t \in \mathbb{R}^N$ represents the battery charging/discharging schedule, depending on the sign of $u_t$ (i.e., positive means charging, and vice versa). The canonical form of the battery dynamics can be written as $x_{t+1} = Ax_t + Bu_t - w_t$, where $A$ is a $N \times N$ matrix which models the self-degradation of the $N$-battery system, $B$ is a $N \times N$ matrix which represents the charging efficiency of each battery unit, $w_t$ is a $N \times 1$ vector which denotes the current demand in terms of the charging rate (kW) of all the EVs connected to the charging stations. Assuming that the initial SoC as $x_0$, the goal is to control the batteries to minimize the difference between the current SoC of all batteries and a nominal value $\bar{x}$, plus a charging/discharging cost to account for battery usage [49, 50], which can be expressed mathematically as $\min_{u_1, u_2, \cdots, u_{T+1}} \sum_{t=1}^{T+1} \|x_t - \bar{x}\|^2 + b\|u_t\|^2$.

This problem falls into SOCO based on the reduction framework described in [49]. Specifically, at time step $t + 1$, we can expand $x_{t+1}$ based on the battery dynamics as $x_{t+1} = A^t x_1 + \sum_{j=1}^{t} A^{t-j} Bu_j - \sum_{j=1}^{t} A^{t-j} w_j$, We define the context parameter as $y_t = \bar{x} - A^t x_1 + \sum_{i=1}^{t} A^{t-i} w_i$ and the action as $a_t = \sum_{i=1}^{t} A^{t-i} Bu_i$. Then, assuming an identity matrix $B$ (ignoring charging loss), the optimization problem becomes $\min_{a_1, \cdots, a_T} \|x_1 - \bar{x}\|^2 + b\|u_T\|^2 + \sum_{t=1}^{T} \|a_t - y_t\|^2 + b\|a_t - Aa_{t-1}\|^2$. Given an initial value of $x_1$, this problem can be further simplified and reformulated as

$$\min_{a_1, a_2, \cdots, a_T} \sum_{t=1}^{T} \frac{1}{b} \|a_t - y_t\|^2 + \|a_t - Aa_{t-1}\|^2, \tag{7}$$

which is in a standard SOCO form by considering $y_t$ as the context and $a_t$ as the action at time $t$.

To validate the effectiveness of RCL, we use a public dataset [51] provided by ElaadNL, a Dutch EV charging infrastructure company. We collect a dataset containing transaction records from ElaadNL charging stations in the Netherlands from January to June of 2019. Each transaction record contains the energy demand, transaction start time and charging time. As the data does not specify the details of battery units, we consider the battery units as a single combine battery by summing up the energy demand within each hour to obtain the hourly energy demand.

We use the January to February data as the training dataset, March to April data as the validation dataset for tuning the hyperparameters such as learning rate, and May to June as the testing dataset. We consider each problem instance as one day ($T = 24$ hours, plus an initial action). Thus, a sliding

window of 25 is applied, moving one hour ahead each time, on the raw data to generate 1416 problem instances, where the first demand of each instance is used as the initial action of all the algorithms. We set $b = 10$ and $A = I$ for the cost function in Eqn. (7).

All the algorithms use the same ML architecture, when applicable, with the same initialized weights in our experiments for fair comparison. To be consistent with the literature [52, 53], all the ML models are trained offline. Specifically, we use a recurrent neural network (RNN) model that contains 2 hidden layers, each with 8 neurons, and implement the model using PyTorch. We train the RNN for 140 epochs with a batch size of 50. When the RNN model is trained as a standalone optimizer in a robustification-oblivious manner, the training process takes around 1 minute on a 2020 MacBook Air with 8GB memory and a M1 chipset. When RNN is trained in a robustification-aware manner, it takes around 2 minutes. The testing process is almost instant and takes less than 1 second.

## B.2    Baseline Algorithms

By default, RCL uses a robustification-aware ML model due to the advantage of average cost performance compared to a robustification-oblivious model. We compare RCL with several representative baseline algorithms as summarized below.

• Offline Optimal Oracle (**OPT**): This is the optimal offline algorithm that has all the contextual information and optimally solves the problem.

• Regularized Online Balanced Descent (**ROBD**): ROBD is the state-of-the-art order-optimal online algorithm with the best-known competitive ratio for our SOCO setting [54, 49]. The parameters of ROBD are all optimally set according to [49]. By default, RCL uses ROBD as its expert for robustness.

• Hitting Cost Minimizer (**HitMin**): HitMin is a special instance of ROBD by setting the parameters such that it greedily minimizing the hitting cost at each time. This minimizer can be empirically effective and hence also used in ROBD as a regularizer.

• Machine Learning Only (**ML**): ML is trained as a standalone optimizer in a robustification-oblivious manner. It does not use robustification during online optimization.

• Expert-Calibrated Learning (**EC-L2O**): It is an ML-augmented algorithm that applies to our SOCO setting by using an ML model to regularize online actions without robustness guarantees [55]. We set its parameters based on the validation dataset to have the optimal average performance with an empirical competitive ratio less than $(1 + \lambda)CR^{\pi}$.

• RCL with a robustification-oblivious ML model (RCL-O): To differentiate the two forms of RCL, we use RCL to refer to RCL with a robustification-aware ML model and RCL-O for the robustification-oblivious ML model, where "-O" represents robustification-obliviousness.

To highlight our key contribution to the SOCO literature, the baseline algorithms we choose are representative of the state-of-the-art expert algorithms, effective heuristics, and ML-augmented algorithms for the SOCO setting we consider. While there are a few other ML-augmented algorithms for SOCO [56, 57, 58], they do not apply to our problem as they consider unsquared switching costs in a metric space and exploit the natural triangular inequality. Adapting them to the squared switching costs is non-trivial.

## B.3    Additional Empirical Results

In Section 5, we have evaluated RCL using ROBD as the expert online algorithm with $\lambda = 1$. Here we provide additional experiment results to further evaluate the effectiveness of RCL, and the results are organized as follows. First, we change the expert online algorithm from ROBD to HitMin to show the flexibility of RCL. Second, we quantitatively present the effect of parameter $\lambda$ in RCL. Third, we introduce additional out-of-distribution samples to test the robustness of RCL and baselines, in terms of competitive ratio. Finally, we experiment under the delayed-feedback setting, which is a more challenging problem setup.

### B.3.1    Utilizing HitMin as the expert

RCL is flexible and can work with any expert online algorithm, even an expert that does not have good or bounded competitive ratios. Thus, it is interesting to see how RCL performs given an alternative

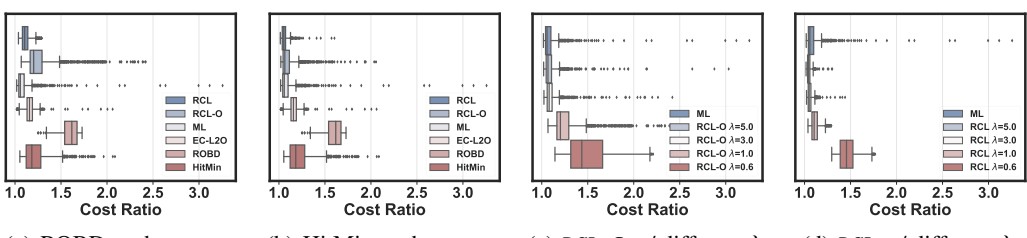

(a) ROBD as the expert    (b) HitMin as the expert    (c) `RCL-O` w/ different $\lambda$    (d) `RCL` w/ different $\lambda$

Figure 2: Cost ratio distributions ($\lambda = 1$ by default).

expert. For example, in Table 1, HitMin empirically outperforms ROBD in terms of the average, although it is not as robust as ROBD. Thus, using $\lambda = 1$, we leverage HitMin as the expert for `RCL` and `RCL-O`, and show the cost ratio distributions in Fig. 2(b). Comparing Fig. 2(b) with Fig. 2(a), we see that `RCL` and `RCL-O` both have many low cost ratios by using HitMin as the expert, but the worst case for `RCL` is not as good as when using ROBD as the expert. For example, the average cost and competitive ratio are 1.0515 and 1.6035, respectively, for `RCL`. This result is not surprising, as the new expert HitMin has a better average performance but worse competitive ratio than the default expert ROBD.

### B.3.2 Impact of $\lambda$

Theorem 4.1 shows the point that we need to set a large enough $\lambda$ in order to provide enough flexibility for `RCL` to exploit good ML predictions. With a small $\lambda > 0$, despite the stronger competitiveness against the expert, it is possible that `RCL` may even empirically perform worse than both the ML model and the expert. Thus, we now investigate the impact of $\lambda$.

We see from Table 1 that the empirical average cost and competitive ratio of `RCL` are both worse with $\lambda = 0.6$ than with the default $\lambda = 1$. More interestingly, by setting $\lambda = 5$, the average cost of `RCL` is even lower than that of ML. This is because ML in our experiment performs fairly well on average. Thus, by setting a large $\lambda = 5$, `RCL` is able to exploit the benefits of good ML predictions for many typical cases, while using the expert ROBD as a safeguard to handle a few bad problem instances for which ML cannot perform well. Also, the empirical competitive ratio of `RCL` is better with $\lambda = 5$ than with $\lambda = 3$, supporting Theorem 4.1 that a larger $\lambda$ may not necessarily increase the competitive ratio as `RCL` can exploit good ML predictions. In addition, given each $\lambda$, `RCL` outperforms `RCL-O`, which highlights the importance of training the ML model in a robustification-aware manner to avoid the mismatch between training and testing objectives.

We also show in Fig. 2(c) an Fig. 2(d) the cost ratio distributions for `RCL-O` and `RCL`, respectively, under different $\lambda$. The results reaffirm our main Theorem 4.1 as well as the importance of training the ML model in a robustification-aware manner.

Next, we show the bi-competitive cost ratios of `RCL-O` against both the expert ROBD and the ML predictions. We focus on `RCL-O` as its ML model is trained as a standalone optimizer, whereas `RCL` uses a robustification-aware ML model that is not specifically trained to produce good pre-robustification predictions. According to Theorem 4.1, `RCL-O` obtains a potentially better competitiveness against ML but a worse competitive against the expert ROBD when $\lambda$ increases, and vice versa. To further validate the theoretical analysis, we test `RCL-O` with different $\lambda$ and obtain the 2D histogram of its bi-competitive cost ratios against ROBD and ML, respectively. The results are shown in Fig. 3. In agreement with our analysis, the cost ratio of `RCL-O` against ROBD never exceeds $(1 + \lambda)$ for any $\lambda > 0$. Also, with a small $\lambda = 0.6$, the cost ratio of `RCL-O` against ROBD concentrates around 1, while it does not exploit the benefits of ML predictions very well. On the other hand, with a large $\lambda = 5$, the cost ratio of `RCL-O` against ROBD can be quite high, although it follows (good) ML predictions more closely for better average performance. Most importantly, by increasing $\lambda > 0$, we can see the general trend that `RCL-O` follows the ML predictions more closely while still being able to guarantee competitiveness against ROBD. Again, this confirms the key point of our main insights in Theorem 4.1.

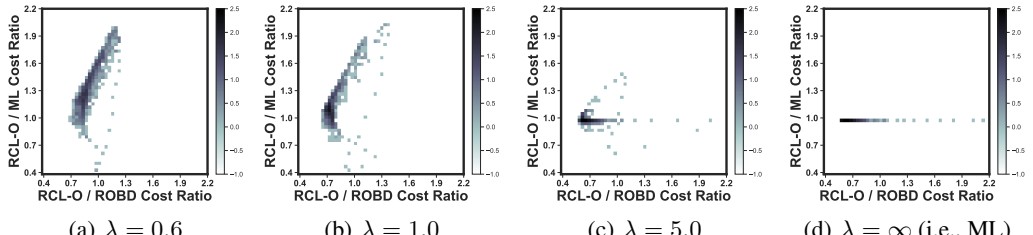

|  |  |  | (a) $\lambda = 0.6$ |  |  | (b) $\lambda = 1.0$ |  |  | (c) $\lambda = 5.0$ |  | (d) $\lambda = \infty$ (i.e., ML) |

Figure 3: Histogram of bi-competitive cost ratios of RCL-O (against ROBD and ML) under different $\lambda$. For better visualization, the color map represents logarithmic values of the cost ratio histogram with a base of 10.

|  |  | $p_c$=0.05 | | | $p_c$=0.1 | | | $p_c$=0.2 | | |
|---|---|---|---|---|---|---|---|---|---|---|
|  |  | $\sigma$ =0.06 | $\sigma$ =0.08 | $\sigma$ =0.1 | $\sigma$ =0.06 | $\sigma$ =0.08 | $\sigma$ =0.1 | $\sigma$ =0.06 | $\sigma$ =0.08 | $\sigma$ =0.1 |
| AVG | RCL | 1.1331 | 1.1444 | 1.1556 | 1.1487 | 1.1693 | 1.1904 | 1.1827 | 1.2254 | 1.2697 |
|  | RCL-O | 1.2425 | 1.2436 | 1.2462 | 1.2416 | 1.2434 | 1.2478 | 1.2370 | 1.2394 | 1.2469 |
|  | **ML** | 1.0722 | 1.0778 | 1.0855 | 1.0770 | 1.0874 | 1.1018 | 1.0858 | 1.1053 | 1.1325 |
|  | **EC-L2O** | 1.1728 | 1.1737 | 1.1754 | 1.1731 | 1.1750 | 1.1784 | 1.1727 | 1.1757 | 1.1815 |
|  | **ROBD** | 1.6048 | 1.6048 | 1.6048 | 1.6048 | 1.6049 | 1.6049 | 1.6048 | 1.6048 | 1.6049 |
|  | **HitMin** | 1.2112 | 1.2195 | 1.2302 | 1.2202 | 1.2357 | 1.2557 | 1.2410 | 1.2724 | 1.3127 |
| CR | RCL | 2.5028 | 2.9697 | 3.2247 | 2.6553 | 3.0283 | 3.2711 | 2.5714 | 3.0123 | 3.1653 |
|  | RCL-O | 2.4209 | 2.4209 | 2.4209 | 2.4209 | 2.4209 | 2.4209 | 2.4209 | 2.4209 | 2.4209 |
|  | **ML** | 6.5159 | 8.9245 | 11.6627 | 4.4025 | 6.5090 | 9.4168 | 5.5798 | 7.3956 | 9.3903 |
|  | **EC-L2O** | 3.4639 | 4.6034 | 5.9666 | 2.6545 | 3.6740 | 5.1129 | 2.9766 | 3.7713 | 4.6983 |
|  | **ROBD** | 1.7291 | 1.7291 | 1.7291 | 1.7291 | 1.7291 | 1.7291 | 1.7291 | 1.7296 | 1.7298 |
|  | **HitMin** | 4.8573 | 6.7746 | 8.8383 | 3.1492 | 4.8253 | 7.0405 | 5.0632 | 6.9699 | 8.9246 |

Table 2: Average cost and competitive ratio comparison of different algorithms. We study the effect of introducing out-of-distribution (OOD) samples. Within the testing dataset, we randomly select a fraction of $p_c$ of samples and add some random noise following $\mathcal{N}(0, \sigma)$ to contaminate these data samples (whose input values are all normalized within $[0, 1]$).

### B.3.3 Larger distributional shifts

In our dataset, ML performs very well on average as the testing distribution matches well with its training distribution. To consider more challenging cases as a stress test, we manually increase the testing distributional shifts by adding random noise following $\mathcal{N}(0, \sigma)$ to a certain faction $p_c$ of the testing samples. Note that, as we intentionally stress test RCL and RCL-O under a larger distributional shift, their ML models remain unchanged as in the default setting and are not re-trained by adding noisy data to the training dataset.

With the default $\lambda = 1$, we show the average cost and competitive ratio results in Table 2. We see that ROBD is very robust and little affected by the distributional shifts. In terms of the competitive ratio, ML, HitMin and EC-L2O are not robust, resulting in a large competitive ratio when we add more noisy samples. The average cost performance of RCL is empirically better than that of RCL-O in almost all cases, except for a slight increase in the practically very rare case where 20% samples are contaminated with large noise. On the other hand, as expected, the competitive ratios of RCL and RCL-O both increase as we add more noise. While RCL has a higher competitive ratio than RCL-O empirically in the experiment, they both have the same guaranteed $(1 + \lambda)$ competitiveness against ROBD regardless of how their ML models are trained. Also, their competitive ratios are both better than other algorithms, showing the effectiveness of our novel robustification process.

### B.4 Results with Feedback Delay

We now turn to the case when there is a one-step feedback delay, i.e., the context parameter $y_t$ is not known to the agent until time $t + 1$. For this setting, we consider the best-known online algorithm iROBD [49] as the expert that handles the feedback delay with a guaranteed competitive ratio with respect to OPT. The other baseline online algorithms — ROBD, EC-L2O, and HitMin— presented in Section B.2 require the immediate revelation of $y_t$ without feedback delay and hence do not directly apply to this case. Thus, for comparison, we use the predicted context, denoted by $\hat{y}_t$, with up to 15% prediction errors in the baseline online algorithms, and reuse the algorithm names (e.g., EC-L2O uses predicted $\hat{y}_t$ as if it were the true context for decision making). We train ML using the same

| | RCL | | | | RCL-O | | | | ML | EC-L2O | iROBD | HitMin | ROBD |
|---|---|---|---|---|---|---|---|---|---|---|---|---|---|
| | λ=0.6 | λ=1 | λ=3 | λ=5 | λ=0.6 | λ=1 | λ=3 | λ=5 | | | | | |
| **AVG** | 1.5011 | 1.3594 | 1.2874 | 1.2899 | 1.5134 | 1.3690 | 1.2949 | 1.3026 | **1.2792** | 1.4112 | 2.3076 | 2.6095 | 2.5974 |
| **CR** | 2.9797 | **2.4832** | 3.2049 | 3.9847 | 2.9797 | 2.4832 | 3.3367 | 4.3040 | 8.4200 | 15.1928 | 4.7632 | 26.0264 | 2.8478 |

Table 3: Competitive ratio and average cost comparison of different algorithms with feedback delay.

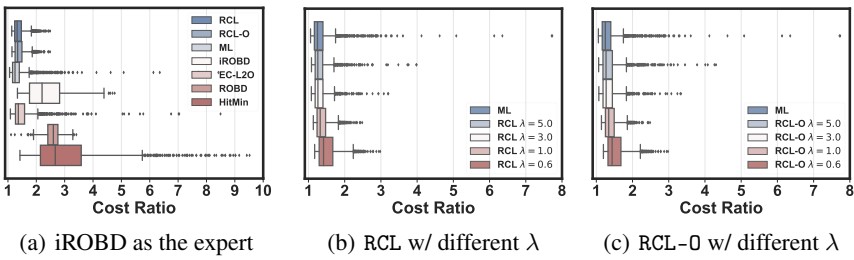

(a) iROBD as the expert     (b) RCL w/ different $\lambda$     (c) RCL-O w/ different $\lambda$

Figure 4: Cost ratio distributions with feedback delay ($\lambda = 1$ by default)

architecture as in Section B.3, with the exception that only delayed context is provided as input for both training and testing. The reported values are normalized with respect to those of the respective offline optimal algorithm OPT. The average cost (**AVG**) and competitive ratio (**CR**) are all empirical results reported on the testing dataset.

We show the results in Table 3 and Fig. 4. We see that with the default $\lambda = 1$, both RCL and RCL-O have a good average cost, but RCL has a lower average cost than RCL-O and is outperformed only by ML in terms of the average cost. RCL and RCL-O have the same competitive ratio guarantee (i.e., $(1 + \lambda)$ times the competitive ratio of iROBD). Nonetheless, RCL has the lowest competitive ratio than all the other algorithms, demonstrating the power of RCL to leverage both ML prediction and the robust expert. In this experiment, both RCL and RCL-O outperform iROBD in terms of the empirical competitive ratio because they are able to exploit the good ML predictions for those problem instances that are difficult for iROBD.

By comparison, ML performs well on average by exploiting the historical data, but has a high competitive ratio. The alternative baselines — ROBD, EC-L2O and HitMin— use predicted context $\hat{y}_t$ as the true context. Except for the good empirical competitive ratio of ROBD, they do not have good average performance or guaranteed robustness due to their naively trusting the predicted context (that can potentially have large prediction errors). Note that the empirical competitive ratio of ROBD with predicted context is still much higher than that with the true context in Table 1. These results reinforce the point that blindly using ML predictions (i.e., predicted context in this example) without additional robustification can lead to poor performance in terms of both average cost and worst-case cost ratio.

We further show in Fig. 4 the box plots for cost ratios of different algorithms, providing a detailed view of the algorithms' performance. The key message is that RCL obtains the best of both worlds — a good average cost and a good competitive ratio. Moreover, we see that by setting $\lambda = 1$, we provide enough freedom to RCL to exploit the benefits of ML predictions while also ensuring worst-case robustness. Thus, like in the no-delay case in Table 1 and Fig. 2, the empirical competitive ratio of RCL with $\lambda = 1$ is even lower than that with $\lambda = 0.6$.

## C   Proof of Theorems and Corollaries in Section 4

### C.1   Proof of Theorem 4.1 (Cost Ratio)

To prove Theorem 4.1, we first give some technical lemmas about the smoothness of cost functions from Lemma C.1 to Lemma C.3.

**Lemma C.1** (Lemma 4 in [59]). *Assume $f(x)$ is $\beta$ smooth, for any $\lambda > 0$, we have*

$$f(x) \leq (1 + \lambda)f(y) + (1 + \frac{1}{\lambda})\frac{\beta}{2}\|x - y\|^2 \quad \forall x, y \in \mathcal{X}$$

**Lemma C.2.** *Assume $f(x)$ is $\beta_1$ smooth and $d(x)$ is $\beta_2$ smooth, then $f(x) + d(x)$ is $\beta_1 + \beta_2$ smooth.*

**Lemma C.3.** *Suppose that the hitting cost $f(x, y_t)$ is $\beta_h$-smooth with respect to $x$, The switching cost is $d(x_t, x_{t-1}) = \frac{1}{2}\|x_t - \delta(x_{t-p:t-1})\|^2$, where $\delta(\cdot)$ is $L_i$-Lipschitz with respect to $x_{t-i}$. Then for any two action sequences $x_{1:T}$ and $x'_{1:T}$, we must have*

$$\text{cost}(x_{1:T}) - (1+\lambda)\text{cost}(x'_{1:T}) \leq \frac{\beta + (1 + \sum_{k=1}^{p} L_k)^2}{2}(1 + \frac{1}{\lambda})\|x_{1:T} - x'_{1:T}\|^2, \quad \forall \lambda > 0 \quad (8)$$

*Proof.* The objective to be bounded can be decomposed as

$$\text{cost}(x_{1:T}) - (1+\lambda)\text{cost}(x'_{1:T})$$
$$= \left(\sum_{t=1}^{T} f(x_t, y_t) - (1+\lambda)f(x'_t, y_t)\right) + \frac{1}{2}\left(\sum_{t=1}^{T} \|x_t - \delta(x_{t-p:t-1})\|^2 - (1+\lambda)\|x'_t - \delta(x'_{t-p:t-1})\|^2\right)$$
$$(9)$$

Since hitting cost is $\beta_h$-smooth, then

$$\sum_{t=1}^{T} f(x_t, y_t) - (1+\lambda)f(x'_t, y_t) \leq \frac{\beta_h}{2}(1 + \frac{1}{\lambda})\sum_{t=1}^{T} \|x_t - x'_t\|^2 \quad (10)$$

Besides, based on the Lipschitz assumption of function $\delta(\cdot)$, we have

$$\|x_t - \delta(x_{t-p:t-1})\|^2 - (1+\lambda)\|x'_t - \delta(x'_{t-p:t-1})\|^2$$
$$\leq (1 + \frac{1}{\lambda})\|(x_t - x'_t) + (\delta(x_{t-p:t-1}) - \delta(x'_{t-p:t-1}))\|^2$$
$$\leq (1 + \frac{1}{\lambda})\left(\|x_t - x'_t\| + \|\delta(x_{t-p:t-1}) - \delta(x'_{t-p:t-1})\|\right)^2$$
$$\leq (1 + \frac{1}{\lambda})\left(\|x_t - x'_t\| + \sum_{k=1}^{p} L_k\|x_{t-k} - x'_{t-k}\|\right)^2 \quad (11)$$
$$\leq (1 + \frac{1}{\lambda})(1 + \sum_{k=1}^{p} L_k)\left(\|x_t - x'_t\|^2 + \sum_{k=1}^{p} L_k\|x_{t-k} - x'_{t-k}\|^2\right)$$

Summing up the switching costs of all time steps together, we have

$$\sum_{t=1}^{T} \|x_t - \delta(x_{t-p:t-1})\|^2 - (1+\lambda)\|x'_t - \delta(x'_{t-p:t-1})\|^2$$
$$\leq (1 + \frac{1}{\lambda})(1 + \sum_{k=1}^{p} L_k)\sum_{t=1}^{T}\left(\|x_t - x'_t\|^2 + \sum_{k=1}^{p} L_k\|x_{t-k} - x'_{t-k}\|^2\right)$$
$$\leq (1 + \frac{1}{\lambda})(1 + \sum_{k=1}^{p} L_k)\sum_{t=1}^{T}(1 + \sum_{k=1}^{p} L_k)\|x_t - x'_t\|^2 \quad (12)$$
$$= (1 + \frac{1}{\lambda})(1 + \sum_{k=1}^{p} L_k)^2 \sum_{t=1}^{T} \|x_t - x'_t\|^2$$

Substituting Eqn. (12) and Eqn. (10) into Eqn. (9), we finish the proof. $\qquad\square$

Now we propose Lemma C.4 based on these above lemmas, which ensures the feasibility of robustness constraint in Eqn. (1)

**Lemma C.4.** *Let $\pi$ be any expert algorithm for the SOCO problem with multi-step feedback delays and multi-step switching costs, for any $\lambda \geq 0$ and $\lambda \geq \lambda_0 \geq 0$, the total cost by the projected actions $x_t$ must satisfy $\text{cost}(x_{1:T}) \leq (1+\lambda)\text{cost}(x^\pi_{1:T})$*

*Proof.* We prove by induction that the constraints in Eqn. (1) are satisfied for each $t$. For $t = 1$, since we assume the initial actions are the same ($x_{-p+1:0} = x^\pi_{-p+1:0}$), it is obvious that $x = x^\pi_1$ satisfies the robustness constraints Eqn. (1).

Then for any time step $t \geq 2$, suppose it holds at $t-1$ that

$$\sum_{\tau \in \mathcal{A}_{t-1}} f(x_\tau, y_\tau) + \sum_{\tau \in \mathcal{A}_{t-1} \cup \mathcal{B}_{t-1}} d(x_\tau, x_{\tau-p:\tau-1}) + \sum_{\tau \in \mathcal{B}_{t-1}} H(x_\tau, x_\tau^\pi) + G(x, x_{t-p:t-1}, x_{t-p:t}^\pi)$$

$$\leq (1+\lambda)\left(\sum_{\tau \in \mathcal{A}_{t-1}} f(x_\tau^\pi, y_\tau) + \sum_{\tau \in \mathcal{A}_{t-1} \cup \mathcal{B}_{t-1}} d(x_\tau^\pi, x_{\tau-p:\tau-1}^\pi)\right)$$

(13)

Now the robustness constraints Eqn. (1) is satisfied if we prove $x_t = x_t^\pi$ satisfies the constraints in Eqn. (1) at time step $t$. Since for the sets $\mathcal{A}$ and $\mathcal{B}$, we have

$$(\mathcal{A}_t \cup \mathcal{B}_t) \setminus (\mathcal{A}_{t-1} \cup \mathcal{B}_{t-1}) = \{t\}, \quad \mathcal{A}_{t-1} \subseteq \mathcal{A}_t,$$ (14)

so it holds that

$$\sum_{\tau \in \mathcal{A}_t \cup \mathcal{B}_t} d(x_\tau, x_{\tau-p:\tau-1}) - \sum_{\tau \in \mathcal{A}_{t-1} \cup \mathcal{B}_{t-1}} d(x_\tau, x_{\tau-p:\tau-1}) = d(x_t, x_{t-p:t-1})$$ (15)

By Lemma C.1, we have

$$d(x_t^\pi, x_{t-p:t-1}) - (1+\lambda)d(x_t^\pi, x_{t-p:t-1}^\pi)$$
$$\leq \frac{1}{2}(1 + \frac{1}{\lambda})\|\delta(x_{t-p:t-1}) - \delta(x_{t-p:t-1}^\pi)\|^2$$
$$\leq \frac{1}{2}(1 + \frac{1}{\lambda})\left(\sum_{i=1}^p L_i\|x_{t-i} - x_{t-i}^\pi\|\right)^2$$

(16)

Denote $\alpha = 1 + \sum_{k=1}^p L_k$. For the reservation cost, we have

$$G(x_{t-1}, x_{t-p-1:t-2}, x_{t-p-1:t-1}^\pi) - G(x_t^\pi, x_{t-p:t-1}, x_{t-p:t}^\pi)$$
$$= \frac{\alpha(1+\frac{1}{\lambda_0})}{2}\left(\sum_{k=1}^p \sum_{i=0}^{p-k} L_{k+i}\|x_{t-i-1} - x_{t-i-1}^\pi\|^2 - \sum_{k=1}^p \sum_{i=1}^{p-k} L_{k+i}\|x_{t-i} - x_{t-i}^\pi\|^2\right)$$
$$= \frac{\alpha(1+\frac{1}{\lambda_0})}{2}\left(\sum_{k=0}^{p-1} \sum_{i=1}^{p-k} L_{k+i}\|x_{t-i} - x_{t-i}^\pi\|^2 - \sum_{k=1}^p \sum_{i=1}^{p-k} L_{k+i}\|x_{t-i} - x_{t-i}^\pi\|^2\right)$$
$$= \frac{\alpha(1+\frac{1}{\lambda_0})}{2}\left(\sum_{k=0}^{p-1} \sum_{i=1}^{p-k} L_{k+i}\|x_{t-i} - x_{t-i}^\pi\|^2 - \sum_{k=1}^{p-1} \sum_{i=1}^{p-k} L_{k+i}\|x_{t-i} - x_{t-i}^\pi\|^2\right)$$
$$= \frac{\alpha(1+\frac{1}{\lambda_0})}{2}\sum_{i=1}^p L_i\|x_{t-i} - x_{t-i}^\pi\|^2$$

(17)

Continuing with Eqn. (17), we have

$$G(x_{t-1}, x_{t-p-1:t-2}, x_{t-p-1:t-1}^\pi) - G(x_t^\pi, x_{t-p:t-1}, x_{t-p:t}^\pi) = \frac{\alpha(1+\frac{1}{\lambda_0})}{2}\sum_{i=1}^p L_i\|x_{t-i} - x_{t-i}^\pi\|^2$$

$$\geq \frac{(1+\frac{1}{\lambda_0})(\sum_{i=1}^p L_i)^2}{2}\sum_{i=1}^p \frac{L_i}{\sum_{i=1}^p L_i}\|x_{t-i} - x_{t-i}^\pi\|^2$$

$$\geq \frac{(1+\frac{1}{\lambda_0})(\sum_{i=1}^p L_i)^2}{2}\left(\sum_{i=1}^p \frac{L_i}{\sum_{i=1}^p L_i}\|x_{t-i} - x_{t-i}^\pi\|\right)^2$$

$$= \frac{1}{2}(1 + \frac{1}{\lambda_0})\left(\sum_{i=1}^p L_i\|x_{t-i} - x_{t-i}^\pi\|\right)^2 \geq \frac{1}{2}(1 + \frac{1}{\lambda})\left(\sum_{i=1}^p L_i\|x_{t-i} - x_{t-i}^\pi\|\right)^2$$

(18)

where the second inequality holds by Jensen's inequality. Therefore, combining with (16), we have

$$d(x_t^\pi, x_{t-p:t-1}) + G(x_t^\pi, x_{t-p:t-1}, x_{t-p:t}^\pi) \leq G(x_{t-1}, x_{t-p-1:t-2}, x_{t-p-1:t-1}^\pi) + (1+\lambda)d(x_t^\pi, x_{t-p:t-1}^\pi)$$
(19)

By Eqn. (19), we have

$$G(x_t^\pi, x_{t-p:t-1}, x_{t-p:t}^\pi) + \sum_{\tau \in \mathcal{A}_t \cup \mathcal{B}_t} d(x_\tau, x_{\tau-p:\tau-1}) - \sum_{\tau \in \mathcal{A}_{t-1} \cup \mathcal{B}_{t-1}} d(x_\tau, x_{\tau-p:\tau-1})$$

$$\leq G(x_{t-1}, x_{t-p-1:t-2}, x_{t-p-1:t-1}^\pi) + (1+\lambda) \left( \sum_{\tau \in \mathcal{A}_t \cup \mathcal{B}_t} d(x_\tau^\pi, x_{\tau-p:\tau-1}^\pi) - \sum_{\tau \in \mathcal{A}_{t-1} \cup \mathcal{B}_{t-1}} d(x_\tau^\pi, x_{\tau-p:\tau-1}^\pi) \right)$$

$$(20)$$

Now we define a new set $\mathcal{D}_t = \mathcal{A}_t \backslash \mathcal{A}_{t-1}$, which denotes the timestep set for the newly received context parameters at $t$.

**Case 1**: If $t \in \mathcal{D}_t$, then $\mathcal{B}_{t-1} \backslash \mathcal{B}_t = \mathcal{D}_t \backslash \{t\}$, then we have

$$\left( \sum_{\tau \in \mathcal{A}_t} f(x_\tau, y_\tau) + \sum_{\tau \in \mathcal{B}_t} H(x_\tau, x_\tau^\pi) \right) - \left( \sum_{\tau \in \mathcal{A}_{t-1}} f(x_\tau, y_\tau) + \sum_{\tau \in \mathcal{B}_{t-1}} H(x_\tau, x_\tau^\pi) \right)$$

$$= \sum_{\tau \in \mathcal{D}_t} f(x_\tau, y_\tau) - \sum_{\tau \in \mathcal{D}_t \backslash \{t\}} H(x_\tau, x_\tau^\pi) = f(x_t^\pi, y_t) + \sum_{\tau \in \mathcal{D}_t \backslash \{t\}} f(x_\tau, y_\tau) - \sum_{\tau \in \mathcal{D}_t \backslash \{t\}} H(x_\tau, x_\tau^\pi)$$

$$(21)$$

Since hitting cost $f(\cdot, y_t)$ is $\beta_h$-smooth, we have

$$\sum_{\tau \in \mathcal{D}_t \backslash \{t\}} f(x_\tau, y_\tau) - \sum_{\tau \in \mathcal{D}_t \backslash \{t\}} (1+\lambda) f(x_\tau^\pi, y_\tau)$$

$$\leq \frac{\beta_h(1+\frac{1}{\lambda})}{2} \sum_{\tau \in \mathcal{D}_t \backslash \{t\}} \|x_\tau^\pi - x_\tau\|^2 \leq \sum_{\tau \in \mathcal{D}_t \backslash \{t\}} H(x_\tau, x_\tau^\pi)$$

$$(22)$$

Substituting Eqn. (22) back to Eqn. (21), we have

$$\left( \sum_{\tau \in \mathcal{A}_t} f(x_\tau, y_\tau) + \sum_{\tau \in \mathcal{B}_t} H(x_\tau, x_\tau^\pi) \right) - \left( \sum_{\tau \in \mathcal{A}_{t-1}} f(x_\tau, y_\tau) + \sum_{\tau \in \mathcal{B}_{t-1}} H(x_\tau, x_\tau^\pi) \right)$$

$$\leq (1+\lambda) \left( \sum_{\tau \in \mathcal{A}_t} f(x_\tau, y_\tau) - \sum_{\tau \in \mathcal{A}_{t-1}} f(x_\tau, y_\tau) \right)$$

$$(23)$$

**Case 2**: If $t \notin \mathcal{D}_t$, then $(\mathcal{B}_{t-1} \cup \{t\}) \backslash \mathcal{B}_t = \mathcal{D}_t$ and we have

$$\left( \sum_{\tau \in \mathcal{A}_t} f(x_\tau, y_\tau) + \sum_{\tau \in \mathcal{B}_t} H(x_\tau, x_\tau^\pi) \right) - \left( \sum_{\tau \in \mathcal{A}_{t-1}} f(x_\tau, y_\tau) + \sum_{\tau \in \mathcal{B}_{t-1}} H(x_\tau, x_\tau^\pi) \right)$$

$$= \sum_{\tau \in \mathcal{D}_t} f(x_\tau, y_\tau) - \sum_{\tau \in \mathcal{D}_t} H(x_\tau, x_\tau^\pi) + H(x_t^\pi, x_t^\pi)$$

$$= \sum_{\tau \in \mathcal{D}_t} f(x_\tau, y_\tau) - \sum_{\tau \in \mathcal{D}_t} H(x_\tau, x_\tau^\pi)$$

$$(24)$$

Since hitting cost $f(\cdot, y_t)$ is $\beta_h$-smooth, we have

$$\sum_{\tau \in \mathcal{D}_t} f(x_\tau, y_\tau) - \sum_{\tau \in \mathcal{D}_t} (1+\lambda) f(x_\tau^\pi, y_\tau) \leq \frac{\beta_h(1+\frac{1}{\lambda})}{2} \sum_{\tau \in \mathcal{D}_t} \|x_\tau^\pi - x_\tau\|^2 \leq \sum_{\tau \in \mathcal{D}_t} H(x_\tau, x_\tau^\pi)$$

$$(25)$$

Since $\lambda \geq 0$, we substitute Eqn. (25) back to Eqn. (24), we have the same conclusion as Eqn (23).

Adding Eqn. (13), Eqn. (20) and Eqn. (23) together, we can prove $x = x_t^\pi$ satisfies the constraints in Eqn. (1). At time step $T$, we have

$$\sum_{\tau \in \mathcal{A}_T} f(x_\tau, y_\tau) + \sum_{\tau \in \mathcal{A}_T \cup \mathcal{B}_T} d(x_\tau, x_{\tau-p:\tau-1}) + \sum_{\tau \in \mathcal{B}_T} \left( f(x_\tau, y_\tau) - (1+\lambda) f(x_\tau^\pi, y_\tau) \right)$$

$$\leq \sum_{\tau \in \mathcal{A}_T} f(x_\tau, y_\tau) + \sum_{\tau \in \mathcal{A}_T \cup \mathcal{B}_T} d(x_\tau, x_{\tau-p:\tau-1}) + \sum_{\tau \in \mathcal{B}_T} H(x_\tau, x_\tau^\pi) \tag{26}$$

$$\leq (1+\lambda) \left( \sum_{\tau \in \mathcal{A}_T} f(x_\tau^\pi, y_\tau) + \sum_{\tau \in \mathcal{A}_T \cup \mathcal{B}_T} d(x_\tau^\pi, x_{\tau-p:\tau-1}^\pi) \right)$$

In other words

$$\sum_{\tau \in \mathcal{A}_T \cup \mathcal{B}_T} \left( f(x_\tau, y_\tau) + d(x_\tau, x_{\tau-p:\tau-1}) \right) \leq (1+\lambda) \sum_{\tau \in \mathcal{A}_T \cup \mathcal{B}_T} \left( f(x_\tau^\pi, y_\tau) + d(x_\tau, x_{\tau-p:\tau-1}) \right) \tag{27}$$

$\square$

In the next lemma, we bound the difference between the projected action and the ML predictions.

**Lemma C.5.** *Suppose hitting cost is $\beta_h$-smooth, given the expert policy $\pi$, ML predictions $\tilde{x}_{1:T}$, for any $\lambda > 0$ and $\lambda_1 > 0$, the total distance between actual actions $x_{1:T}$ and ML predictions $\tilde{x}_{1:T}$ are bounded,*

$$\sum_{i=1}^{T} \|x_t - \tilde{x}_t\|^2 \leq \sum_{i=1}^{T} \left( \left[ \|\tilde{x}_t - x_t^\pi\| - \sqrt{K \left( d(x_t^\pi, x_{t-p:t-1}^\pi) + \sum_{\tau \in \mathcal{D}_t} f(x_\tau^\pi, y_\tau) \right)} \right]^+ \right)^2 \tag{28}$$

*where $[\cdot]^+$ is the ReLU function and $K = \frac{2(\lambda - \lambda_0)}{\beta_h(1 + \frac{1}{\lambda_0}) + \alpha^2(1 + \frac{1}{\lambda_0})}$, $\alpha = 1 + \sum_{i=1}^{p} L_i$*

*Proof.* Suppose we at $t-1$ have the following inequality:

$$\sum_{\tau \in \mathcal{A}_{t-1}} f(x_\tau, y_\tau) + \sum_{\tau \in \mathcal{A}_{t-1} \cup \mathcal{B}_{t-1}} d(x_\tau, x_{\tau-p:\tau-1}) + \sum_{\tau \in \mathcal{B}_{t-1}} H(x_\tau, x_\tau^\pi) + G(x, x_{t-p:t-1}, x_{t-p:t}^\pi)$$

$$\leq (1+\lambda) \left( \sum_{\tau \in \mathcal{A}_{t-1}} f(x_\tau^\pi, y_\tau) + \sum_{\tau \in \mathcal{A}_{t-1} \cup \mathcal{B}_{t-1}} d(x_\tau^\pi, x_{\tau-p:\tau-1}^\pi) \right) \tag{29}$$

Remember that $\mathcal{D}_t = \mathcal{A}_t \backslash \mathcal{A}_{t-1}$ is the set of the time steps for the newly received context parameters at $t$. The robustness constraint in Eqn. (1) is satisfied if $x_t$ satisfies the following inequality.

$$\left( \sum_{\tau \in \mathcal{D}_t} f(x_\tau, y_\tau) + \sum_{\tau \in \mathcal{B}_t} H(x_\tau, x_\tau^\pi) - \sum_{\tau \in \mathcal{B}_{t-1}} H(x_\tau, x_\tau^\pi) \right) + d(x_t, x_{t-p:t-1}) + G(x_t, x_{t-p:t-1}, x_{t-p:t}^\pi)$$

$$- G(x_{t-1}, x_{t-p-1:t-2}, x_{t-p-1:t-1}^\pi) \leq (1+\lambda) \left( d(x_t^\pi, x_{t-p:t-1}^\pi) + \sum_{\tau \in \mathcal{D}_t} f(x_\tau^\pi, y_\tau) \right) \tag{30}$$

For the switching cost, we have

$$d(x, x_{t-p:t-1}) - (1+\lambda_0) d(x_t^\pi, x_{t-p:t-1}^\pi)$$

$$\leq \frac{1}{2} (1 + \frac{1}{\lambda_0}) \left( \|x - x_t^\pi\| + \|\delta(x_{t-p:t-1}) - \delta(x_{t-p:t-1}^\pi)\| \right)^2$$

$$\leq \frac{1}{2} (1 + \frac{1}{\lambda_0}) \left( \|x - x_t^\pi\| + \sum_{i=1}^{p} L_i \|x_{t-i} - x_{t-i}^\pi\| \right)^2 \tag{31}$$

$$\leq \frac{\alpha(1 + \frac{1}{\lambda_0})}{2} \left( \|x - x_t^\pi\|^2 + \sum_{i=1}^{p} L_i \|x_{t-i} - x_{t-i}^\pi\|^2 \right)$$

The first inequality comes from Lemma C.1, the second inequality comes from the $L_i$-Lipschitz assumption, and the third inequality is because $\alpha \geq 1$. Besides, from Eqn (17), we have

$$G(x_{t-1}, x_{t-p-1:t-2}, x_{t-p-1:t-1}^\pi) - G(x_t^\pi, x_{t-p:t-1}, x_{t-p:t}^\pi) = \frac{\alpha(1 + \frac{1}{\lambda_0})}{2} \sum_{i=1}^{p} L_i \|x_{t-i} - x_{t-i}^\pi\|^2 \tag{32}$$

Thus we have

$$G(x, x_{t-p:t-1}, x_{t-p:t}^\pi) - G(x_{t-1}, x_{t-p-1:t-2}, x_{t-p-1:t-1}^\pi)$$
$$= G(x, x_{t-p:t-1}, x_{t-p:t}^\pi) - G(x_t^\pi, x_{t-p:t-1}, x_{t-p:t}^\pi) + G(x_t^\pi, x_{t-p:t-1}, x_{t-p:t}^\pi) - G(x_{t-1}, x_{t-p-1:t-2}, x_{t-p-1:t-1}^\pi)$$
$$= G(x, x_{t-p:t-1}, x_{t-p:t}^\pi) - G(x_t^\pi, x_{t-p:t-1}, x_{t-p:t}^\pi) - \frac{\alpha(1 + \frac{1}{\lambda_0})}{2} \sum_{i=1}^{p} L_i \|x_{t-i} - x_{t-i}^\pi\|^2. \tag{33}$$

Combining with inequality (31), we have

$$G(x_t, x_{t-p:t-1}, x_{t-p:t}^\pi) - G(x_{t-1}, x_{t-p-1:t-2}, x_{t-p-1:t-1}^\pi) + d(x_t, x_{t-p:t-1}) - (1 + \lambda_0)d(x_t^\pi, x_{t-p:t-1}^\pi)$$

$$\leq G(x_t, x_{t-p:t-1}, x_{t-p:t}^\pi) - G(x_t^\pi, x_{t-p:t-1}, x_{t-p:t}^\pi) + \frac{\alpha(1 + \frac{1}{\lambda_0})}{2} \|x_t - x_t^\pi\|^2$$

$$= \frac{\alpha(1 + \frac{1}{\lambda_0}) \sum_{k=1}^{p} L_k}{2} \|x_t - x_t^\pi\|^2 + \frac{\alpha(1 + \frac{1}{\lambda_0})}{2} \sum_{k=1}^{p} \|x_t - x_t^\pi\|^2$$

$$= \frac{\alpha^2(1 + \frac{1}{\lambda_0})}{2} \|x_t - x_t^\pi\|^2 \tag{34}$$

Substituting Eqn. (34) back to Eqn. (30), we have

$$\sum_{\tau \in \mathcal{D}_t} (f(x_\tau, y_\tau) - (1 + \lambda_0)f(x_\tau^\pi, y_\tau)) + \sum_{\tau \in \mathcal{B}_t} H(x_\tau, x_\tau^\pi) - \sum_{\tau \in \mathcal{B}_{t-1}} H(x_\tau, x_\tau^\pi)$$
$$+ \frac{\alpha^2(1 + \frac{1}{\lambda_0})}{2} \|x - x_t^\pi\|^2 \leq (\lambda - \lambda_0) \left( d(x_t^\pi, x_{t-p:t-1}^\pi) + \sum_{\tau \in \mathcal{D}_t} f(x_\tau^\pi, y_\tau) \right) \tag{35}$$

**Case 1**: If $t \in \mathcal{D}_t$, then $\mathcal{B}_{t-1} \backslash \mathcal{B}_t = \mathcal{D}_t \backslash \{t\}$, then Eqn.(35) becomes

$$f(x_t, y_t) - (1 + \lambda_0)f(x_t^\pi, y_t) + \frac{\alpha^2(1 + \frac{1}{\lambda_0})}{2} \|x - x_t^\pi\|^2$$
$$+ \sum_{\tau \in \mathcal{D}_t \backslash \{t\}} f(x_\tau, y_\tau) - (1 + \lambda_0)f(x_\tau^\pi, y_\tau) - H(x_\tau, x_\tau^\pi) \leq (\lambda - \lambda_0) \left( d(x_t^\pi, x_{t-p:t-1}^\pi) + \sum_{\tau \in \mathcal{D}_t} f(x_\tau^\pi, y_\tau) \right) \tag{36}$$

Since hitting cost is $\beta_h$-smooth, the sufficient condition for Eqn. (35) becomes

$$\frac{(\beta_h + \alpha^2)(1 + \frac{1}{\lambda_0})}{2} \|x - x_t^\pi\|^2 \leq (\lambda - \lambda_0) \left( d(x_t^\pi, x_{t-p:t-1}^\pi) + \sum_{\tau \in \mathcal{D}_t} f(x_\tau^\pi, y_\tau) \right) \tag{37}$$

Since the hitting cost is non-negative, the sufficient condition can be further simplified, which is

$$\frac{(\beta_h + \alpha^2)(1 + \frac{1}{\lambda_0})}{2} \|x - x_t^\pi\|^2 \leq (\lambda - \lambda_0) \left( f(x_t^\pi, y_t) + d(x_t^\pi, x_{t-p:t-1}^\pi) \right) \tag{38}$$

**Case 2**: If $t \notin \mathcal{D}_t$, then $(\mathcal{B}_{t-1} \cup \{t\}) \backslash \mathcal{B}_t = \mathcal{D}_t$, then Eqn.(35) becomes

$$\frac{\alpha^2(1 + \frac{1}{\lambda_0})}{2} \|x - x_t^\pi\|^2 + H(x, x_t^\pi) + \sum_{\tau \in \mathcal{D}_t} (f(x_\tau, y_\tau) - (1 + \lambda_0)f(x_\tau^\pi, y_\tau) - H(x_\tau, x_\tau^\pi))$$

$$\leq (\lambda - \lambda_0) \left( d(x_t^\pi, x_{t-p:t-1}^\pi) + \sum_{\tau \in \mathcal{D}_t} f(x_\tau^\pi, y_\tau) \right) \tag{39}$$

Since hitting cost is $\beta_h$-smooth, the sufficient condition for Eqn. (39) becomes

$$\frac{(\beta_h + \alpha^2)(1 + \frac{1}{\lambda_0})}{2}\|x - x_t^\pi\|^2 \leq (\lambda - \lambda_0)\left(d(x_t^\pi, x_{t-p:t-1}^\pi) + \sum_{\tau \in \mathcal{D}_t} f(x_\tau^\pi, y_\tau)\right) \quad (40)$$

Now we define

$$K = \frac{2(\lambda - \lambda_0)}{(\beta_h + \alpha^2)(1 + \frac{1}{\lambda_0})}$$

At time step $t$, if $x_t'$ is the solution to this alternative optimization problem

$$x_t' = \arg\min_x \frac{1}{2}\|x - \tilde{x}_t\|^2$$

$$s.t. \quad \|x - x_t^\pi\|^2 \leq K\left(d(x_t^\pi, x_{t-p:t-1}^\pi) + \sum_{\tau \in \mathcal{D}_t} f(x_\tau^\pi, y_\tau)\right) \quad (41)$$

The solution to this problem can be calculated asd

$$x_t' = \theta x_t^\pi + (1 - \theta)\tilde{x}_t$$

$$\theta = \left[1 - \frac{\sqrt{K\left(d(x_t^\pi, x_{t-p:t-1}^\pi) + \sum_{\tau \in \mathcal{D}_t} f(x_\tau^\pi, y_\tau)\right)}}{\|\tilde{x}_t - x_t^\pi\|}\right]^+. \quad (42)$$

Then $\|x_t' - \tilde{x}_t\| = \left[\|\tilde{x}_t - x_t^\pi\| - \sqrt{K\left(d(x_t^\pi, x_{t-p:t-1}^\pi) + \sum_{\tau \in \mathcal{D}_t} f(x_\tau^\pi, y_\tau)\right)}\right]^+$. Since $x_t'$ also satisfies the original robustness constraint, we have $\|x_t - \tilde{x}_t\| \leq \|x_t' - \tilde{x}_t\|$ and we finish the proof.

$\square$

**Proof of Theorem 4.1**
Now summing up the distance through 1 to T, we have

$$\sum_{i=1}^{T}\|x_t - \tilde{x}_t\|^2 \leq \sum_{i=1}^{T}\left(\left[\|\tilde{x}_t - x_t^\pi\| - \sqrt{K\left(d(x_t^\pi, x_{t-p:t-1}^\pi) + \sum_{\tau \in \mathcal{D}_t} f(x_\tau^\pi, y_\tau)\right)}\right]^+\right)^2 \quad (43)$$

Based on Lemma C.3 we have $\forall \lambda_2 > 0$,

$$\text{cost}(x_{1:T}) - (1 + \lambda_2)\text{cost}(\tilde{x}_{1:T}) \leq \frac{\beta + \alpha^2}{2}(1 + \frac{1}{\lambda_2})\sum_{i=1}^{T}\|x_t - \tilde{x}_t\|^2. \quad (44)$$

Suppose the offline optimal action sequence is $x_{1:T}^*$, the optimal cost is $\text{cost}(x_{1:T}^*)$. Then we divide both sides of Eqn. (44) by $\text{cost}(x_{1:T}^*)$, and get $\forall \lambda_2 > 0$,

$$\text{cost}(x_{1:T}) \leq (1 + \lambda_2)\text{cost}(\tilde{x}_{1:T}) + \frac{\beta + \alpha^2}{2}(1 + \frac{1}{\lambda_2})\cdot$$

$$\sum_{i=1}^{T}\left(\left[\|\tilde{x}_t - x_t^\pi\| - \sqrt{K\left(d(x_t^\pi, x_{t-p:t-1}^\pi) + \sum_{\tau \in \mathcal{D}_t} f(x_\tau^\pi, y_\tau)\right)}\right]^+\right)^2 \quad (45)$$

By substituting $K = \frac{2(\lambda - \lambda_0)}{(\beta_h + \alpha^2)(1 + \frac{1}{\lambda_0})}$ back to Eqn (46), we have

$$\text{cost}(x_{1:T}) \leq (1 + \lambda_2)\text{cost}(\tilde{x}_{1:T}) + (1 + \frac{1}{\lambda_2})\sum_{i=1}^{T}\left[\frac{\beta + \alpha^2}{2}\|\tilde{x}_t - x_t^\pi\|^2\right.$$

$$\left. - \frac{\lambda - \lambda_0}{1 + \frac{1}{\lambda_0}}\left(d(x_t^\pi, x_{t-p:t-1}^\pi) + \sum_{\tau \in \mathcal{D}_t} f(x_\tau^\pi, y_\tau)\right)\right]^+ \quad (46)$$

By defining single step cost of the expert $\pi$ as $\text{cost}_t^\pi = d(x_t^\pi, x_{t-p:t-1}^\pi) + \sum_{\tau \in \mathcal{D}_t} f(x_\tau^\pi, y_\tau)$ and the auxiliary cost as $\Delta(\lambda) = \sum_{i=1}^T \left[ \|\tilde{x}_t - x_t^\pi\|^2 - \frac{2(\lambda-\lambda_0)}{(\beta_h+\alpha^2)(1+\frac{1}{\lambda_0})}\text{cost}_t^\pi \right]^+$

$$\text{cost}(x_{1:T}) \leq \left( \sqrt{\text{cost}(\tilde{x}_{1:T})} + \sqrt{\frac{\beta+\alpha^2}{2}\Delta(\lambda)} \right)^2 \tag{47}$$

Combined with Lemma C.4, we obtain the following bound, which finished this proof.

$$\text{cost}(x_{1:T}) \leq \min\left( (1+\lambda)\text{cost}(x_{1:T}^\pi), \left( \sqrt{\text{cost}(\tilde{x}_{1:T})} + \sqrt{\frac{\beta+\alpha^2}{2}\Delta(\lambda)} \right)^2 \right) \tag{48}$$

## C.2    Proof of Theorem 4.2

*Proof.* We first give the formal definition of Rademacher complexity of the ML model space with robustification.

**Definition 5** (Rademacher Complexity). *Let $\text{Rob}_\lambda(\mathcal{W}) = \{\text{Rob}_\lambda(h_W), W \in \mathcal{W}\}$ be the ML model space with robustification constrained by (2). Given the dataset $\mathcal{S}$, the Rademacher complexity with respect to $\text{Rob}_\lambda(\mathcal{W})$ is*

$$\text{Rad}_\mathcal{S}(\text{Rob}_\lambda(\mathcal{W})) = \frac{1}{|\mathcal{S}|}\mathbb{E}_\nu \left[ \sup_{W \in \mathcal{W}} \left( \sum_{i \in \mathcal{S}} \nu_i \text{Rob}_\lambda\left( h_W(y^i) \right) \right) \right],$$

*where $y^i$ is the $i$-th sample in $\mathcal{S}$, and $\nu_1, \cdots, \nu_n$ are independently drawn from Rademacher distribution.*

Since the cost functions are smooth, they are locally Lipschitz continuous for the bounded action space, and we can apply the generalization bound based on Rademacher complexity [60] for the space of robustified ML model $\text{Rob}_\lambda(h_W)$. Given any ML model $h_W$ trained on dataset $\mathcal{S}$, with probability at least $1 - \delta, \delta \in (0, 1)$,

$$\mathbb{E}_{\mathbb{P}_y'}[\text{cost}_{1:T}] \leq \overline{\text{cost}}_\mathcal{S}(\text{Rob}_\lambda(h_W)) + 2\Gamma_x\text{Rad}_\mathcal{S}(\text{Rob}_\lambda(\mathcal{W})) + 3\bar{c}\sqrt{\frac{\log(2/\delta)}{|\mathcal{S}|}}, \tag{49}$$

where $\Gamma_x = \sqrt{T}|\mathcal{X}| \left[ \beta_h + \frac{1}{2}(1 + \sum_{i=1}^p L_i)(1 + \sum_{i=1}^p L_i) \right]$ with $|\mathcal{X}|$ being the size of the action space $\mathcal{X}$ and $\beta_h$, $L_i$, and $p$ as the smoothness constant, Lipschitz constant of the nonlinear term in the switching cost, and the memory length as defined in Assumptions 1 and 2, and $\bar{c}$ is the upper bound of the total cost for an episode. We can get the average cost bound in Proposition 4.2.

Next, we prove that the Rademacher complexity of the ML model space with robustification is no larger than the Rademacher complexity of the ML model space without robustification expressed as $\{h_W, W \in \mathcal{W}\}$, i.e. we need to prove $\text{Rad}_\mathcal{S}(\text{Rob}_\lambda(\mathcal{W})) \leq \text{Rad}_\mathcal{S}(\mathcal{W})$. The Rademacher complexity can be expressed by Dudley's entropy integral [61] as

$$\text{Rad}_\mathcal{S}(\text{Rob}_\lambda(\mathcal{W})) = \mathcal{O}\left( \frac{1}{\sqrt{|\mathcal{S}|}} \int_0^\infty \sqrt{\log \mathbb{N}(\epsilon, \text{Rob}_\lambda(\mathcal{W}), L_2(\mathcal{S}))}d\epsilon \right), \tag{50}$$

where $\mathbb{N}(\epsilon, \text{Rob}_\lambda(\mathcal{W}), L_2(\mathcal{S}))$ is the covering number [61] with respect to radius $\epsilon$ and the function distance metric $\|h_1 - h_2\|_{L_2(\mathcal{S})} = \frac{1}{|\mathcal{S}|}\sum_{i \in \mathcal{S}}\|h_1(x_i) - h_2(x_i)\|^2$ where $h_1$ and $h_2$ are two functions defined on the space including dataset $\mathcal{S}$. We can find that for any two different weights $W_1$ and $W_2$, their corresponding post-robustification distance $\|\text{Rob}_\lambda(h_{W_1}) - \text{Rob}_\lambda(h_{W_2})\|_{L_2(\mathcal{S})}$ is no larger than their pre-robustification distance $\|h_{W_1} - h_{W_2}\|_{L_2(\mathcal{S})}$. To see this, we discuss three cases given any input sample $y$. If both $h_{W_1}(y)$ and $h_{W_2}(y)$ lie in the projection set, then $\text{Rob}_\lambda(h_{W_1})(y) = h_{W_1}(y)$ and $\text{Rob}_\lambda(h_{W_2})(y) = h_{W_2}(y)$. If $h_{W_1}(y)$ lies in the projection set while $h_{W_2}(y)$ is out of the projection set, the projection operation based on the closed convex projection set will make $\|\text{Rob}_\lambda(h_{W_1})(y) - \text{Rob}_\lambda(h_{W_2})(y)\|$ to be less than $\|h_{W_1}(y) - h_{W_2}(y)\|$. If both $h_{W_1}(y)$ and $h_{W_2}(y)$ lie out of the projection set, we still have $\|\text{Rob}_\lambda(h_{W_1})(y) - \text{Rob}_\lambda(h_{W_2})(y)\| \leq \|h_{W_1}(y) - h_{W_2}(y)\|$

since the projection set at each round is a closed convex set [62]. Therefore, after robustification, the distance between two models with different weights will not become larger, i.e. $\|\text{Rob}_\lambda(h_{W_1}) - \text{Rob}_\lambda(h_{W_2})\|_{L_2(\mathcal{S})} \leq \|h_{W_1} - h_{W_2}\|_{L_2(\mathcal{S})}$, which means RCL has a covering number $\mathbb{N}(\epsilon, \text{Rob}_\lambda(\mathcal{W}), L_2(\mathcal{S}))$ no larger than that of the individual ML model $\mathbb{N}(\epsilon, \mathcal{W}, L_2(\mathcal{S}))$ for any $\epsilon$. Thus the Rademacher complexity with the robustification procedure does not increase.

By [63], the upper bound of Rademacher complexity with respect to the space of ML model $\text{Rad}_\mathcal{S}(\text{Rob}_\lambda(\mathcal{W}))$ is in the order of $\mathcal{O}(\frac{1}{\sqrt{|\mathcal{S}|}})$. Since the Rademacher complexity with the robustification procedure satisfies $\text{Rad}_\mathcal{S}(\text{Rob}_\lambda(\mathcal{W})) \leq \text{Rad}_\mathcal{S}(\mathcal{W})$, it also decreases with the dataset size in the order of $\mathcal{O}(\frac{1}{\sqrt{|\mathcal{S}|}})$. □

## D  Robustification-aware Training

Theorem 4.2 also shows the benefits of training the ML model in a robustification-aware manner. Specifically, by comparing the losses in (5) and (6), we see that using (6) as the robustification-aware loss for training $W$ can reduce the term $\overline{\text{cost}}_\mathcal{S}(\text{ROB}(h_W))$ in the average cost bound, which matches exactly with the training objective in (6). The robustification-aware approach is only beginning to be explored in the ML-augmented algorithm literature and non-trivial (e.g., unconstrained downstream optimization in [55]), especially considering that (1) is a constrained optimization problem with no explicit gradients.

Gradient-based optimizers such as Adam [64] are the de facto state-of-the-art algorithms for training ML models, offering better optimization results, convergence, and stability compared to those non-gradient-based alternatives [65]. Thus, it is crucial to derive the gradients of the loss function with respect to the ML model weight $W$ given the added robustification step.

Next, we derive the gradients of $x_t$ with respect to $\tilde{x}_t$. For the convenience of presentation, we use the basic SOCO setting with a single-step switching cost and no hitting cost delay as an example, while noting that the same technique can be extended to derive gradients in more general settings. Specifically, for this setting, the pre-robustification prediction is given by $\tilde{x}_t = h_W(\tilde{x}_{t-1}, y_t)$, where $W$ denotes the ML model weight. Then, the actual post-robustification action $x_t$ is obtained by projection in (1) by setting $q = 0$ and $p = 1$, given the ML prediction $\tilde{x}_t$, the expert's action $x_t^\pi$ and cumulative $\text{cost}(x_{1:t}^\pi)$ up to $t$, and the actual cumulative $\text{cost}(x_{1:t-1})$ up to $t - 1$.

The gradient of the loss function $\text{cost}(x_{1:T}) = \sum_{t=1}^T (f(x_t, y_t) + d(x_t, x_{t-1}))$ with respect to the ML model weight $W$ is given by $\sum_{t=1}^T \nabla_W (f(x_t, y_t) + d(x_t, x_{t-1}))$. Next, we write the gradient of per-step cost with with respect to $W$ as follows:

$$\nabla_W \big(f(x_t, y_t) + d(x_t, x_{t-1})\big)$$
$$= \nabla_{x_t}\big(f(x_t, y_t) + d(x_t, x_{t-1})\big)\nabla_W x_t + \nabla_{x_{t-1}}\big(f(x_t, y_t) + d(x_t, x_{t-1})\big)\nabla_W x_{t-1} \quad (51)$$
$$= \nabla_{x_t}\big(f(x_t, y_t) + d(x_t, x_{t-1})\big)\nabla_W x_t + \nabla_{x_{t-1}}d(x_t, x_{t-1})\nabla_W x_{t-1},$$

where the gradients $\nabla_{x_t}\big(f(x_t, y_t) + d(x_t, x_{t-1})\big)$ and $\nabla_{x_{t-1}}d(x_t, x_{t-1})$ are trivial given the hitting and switching cost functions, and the gradient $\nabla_W x_{t-1}$ is obtained at time $t - 1$ in the same way as $\nabla_W x_t$. To derive $\nabla_W x_t$, by the chain rule, we have:

$$\nabla_W x_t = \nabla_{\tilde{x}_t} x_t \nabla_W \tilde{x}_t + \nabla_{\text{cost}(x_{1:t-1})} x_t \nabla_W \text{cost}(x_{1:t-1}), \quad (52)$$

where $\nabla_W \tilde{x}_t$ is the gradient of the ML output (following a recurrent architecture illustrated in Fig. 1 in the appendix) with respect to the weight $W$ and can be obtained recursively by using off-the-shelf BPTT optimizers [64], and $\nabla_W \text{cost}(x_{1:t-1}) = \sum_{\tau=1}^{t-1} \nabla_W \big(f(x_\tau, y_\tau) + d(x_\tau, x_{\tau-1})\big)$ can also be recursively calculated once we have the gradient in Eqn. (51). Nonetheless, it is non-trivial to calculate the two gradient terms in Eqn. (52), i.e., $\nabla_{\tilde{x}_t} x_t$ and $\nabla_{\text{cost}(x_{1:t-1})} x_t$, where $x_t$ itself is the solution to the constrained optimization problem (1) unlike in the simpler unconstrained case [55]. As we cannot explicitly write $x_t$ in a closed form in terms of $\tilde{x}_t$ and $\text{cost}(x_{1:t-1})$, we leverage the KKT conditions [66, 67, 68] to implicitly derive $\nabla_{\tilde{x}_t} x_t$ and $\nabla_{\text{cost}(x_{1:t-1})} x_t$ in the next proposition.

**Proposition D.1** (Gradients by KKT conditions). *Let $x_t \in \mathcal{X}$ and $\mu \geq 0$ be the primal and dual solutions to the problem* (1)*, respectively. The gradients of $x_t$ with respect to $\tilde{x}_t$ and $\text{cost}(x_{1:t-1})$ are*

$$\nabla_{\tilde{x}_t} x_t = \Delta_{11}^{-1}[I + \Delta_{12}Sc(\Delta, \Delta_{11})^{-1}\Delta_{21}\Delta_{11}^{-1}],$$

$$\nabla_{cost(x_{1:t-1})} x_t = \Delta_{11}^{-1} \Delta_{12} Sc(\Delta, \Delta_{11})^{-1} \mu,$$

*where* $\Delta_{11} = I + \mu\left(\nabla_{x_t,x_t} f(x_t, y_t) + \left(1 + (1 + \frac{1}{\lambda_0})(L_1^2 + L_1)\right) I\right)$, $\Delta_{12} = \nabla_{x_t} f(x_t, y_t) + (x_t - \delta(x_{t-1})) + \left(1 + (1 + \frac{1}{\lambda_0})(L_1^2 + L_1)\right)(x_t - x_t^\pi)$, $\Delta_{21} = \mu \Delta_{12}^\top$, $\Delta_{22} = f(x_t, y_t) + d(x_t, x_{t-1}) + G(x_t, x_t^\pi) + cost(x_{1:t-1}) - (1 + \lambda)cost(x_{1:t}^\pi)$, *and* $Sc(\Delta, \Delta_{11}) = \Delta_{22} - \Delta_{21}\Delta_{11}^{-1}\Delta_{12}$ *is the Schur-complement of* $\Delta_{11}$ *in the blocked matrix* $\Delta = \left[[\Delta_{11}, \Delta_{12}], [\Delta_{21}, \Delta_{22}]\right]$.

If the ML prediction $\tilde{x}_t$ happens to lie on the boundary such that the inequality in (1) becomes an equality for $x = \tilde{x}_t$, then the gradient does not exist in this case and $Sc(\Delta, \Delta_{11})$ may not be full-rank. Nonetheless, we can still calculate the pseudo-inverse of $Sc(\Delta, \Delta_{11})$ and use Proposition D.1 to calculate the subgradient. Such approximation is actually a common practice to address non-differentiable points for training ML models, e.g., using $0$ as the subgradient of $ReLu(\cdot)$ at the zero point [64].

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
