$$+ \frac{\alpha^2(1 + \frac{1}{\lambda_0})}{2}\|x - x^{\pi}_t\|^2 \leq (\lambda - \lambda_0)\left(d(x^{\pi}_t, x^{\pi}_{t-p:t-1}) + \sum_{\tau \in \mathcal{D}_t}f(x^{\pi}_\tau, y_\tau)\right)$$

$$\tag{35}$$

**Case 1**: If $t \in \mathcal{D}_t$, then $\mathcal{B}_{t-1}\backslash\mathcal{B}_t = \mathcal{D}_t\backslash\{t\}$, then Eqn.(35) becomes

$$f(x_t, y_t) - (1 + \lambda_0)f(x^{\pi}_t, y_t) + \frac{\alpha^2(1 + \frac{1}{\lambda_0})}{2}\|x - x^{\pi}_t\|^2$$

$$+ \sum_{\tau \in \mathcal{D}_t\backslash\{t\}}f(x_\tau, y_\tau) - (1 + \lambda_0)f(x^{\pi}_\tau, y_\tau) - H(x_\tau, x^{\pi}_\tau) \leq (\lambda - \lambda_0)\left(d(x^{\pi}_t, x^{\pi}_{t-p:t-1}) + \sum_{\tau \in \mathcal{D}_t}f(x^{\pi}_\tau, y_\tau)\right)$$

$$\tag{36}$$

Since hitting cost is $\beta_h$-smooth, the sufficient condition for Eqn. (35) becomes

$$\frac{(\beta_h + \alpha^2)(1 + \frac{1}{\lambda_0})}{2}\|x - x^{\pi}_t\|^2 \leq (\lambda - \lambda_0)\left(d(x^{\pi}_t, x^{\pi}_{t-p:t-1}) + \sum_{\tau \in \mathcal{D}_t}f(x^{\pi}_\tau, y_\tau)\right) \tag{37}$$

Since the hitting cost is non-negative, the sufficient condition can be further simplified, which is

$$\frac{(\beta_h + \alpha^2)(1 + \frac{1}{\lambda_0})}{2}\|x - x^{\pi}_t\|^2 \leq (\lambda - \lambda_0)\left(f(x^{\pi}_t, y_t) + d(x^{\pi}_t, x^{\pi}_{t-p:t-1})\right) \tag{38}$$

**Case 2**: If $t \notin \mathcal{D}_t$, then $(\mathcal{B}_{t-1} \cup \{t\})\backslash\mathcal{B}_t = \mathcal{D}_t$, then Eqn.(35) becomes

$$\frac{\alpha^2(1 + \frac{1}{\lambda_0})}{2}\|x - x^{\pi}_t\|^2 + H(x, x^{\pi}_t) + \sum_{\tau \in \mathcal{D}_t}(f(x_\tau, y_\tau) - (1 + \lambda_0)f(x^{\pi}_\tau, y_\tau) - H(x_\tau, x^{\pi}_\tau))$$