# OpenReview forum: "Robust Learning for Smoothed Online Convex Optimization with Feedback Delay"
_NeurIPS.cc/2023/Conference — NeurIPS 2023 poster_

### Official Review · Reviewer_atug · 2023-06-12

**Soundness:** 3 good
**Presentation:** 2 fair
**Contribution:** 3 good
**Rating:** 6
**Confidence:** 3

**Summary:**

This paper studies Smoothed Online Convex Optimization (SOCO) problems where the target is to minimize the sum of a per-round hitting cost and a switching cost that penalizes temporal changes in actions.

The most significant difference from previous studies is that they assume a delayed hitting cost and a nonlinear multi-step switching cost, which is quite more general than before and thus brings technical difficulties to provide theoretically guaranteed methods.
Furthermore, they also want to use ML-based online optimizers to enhance performance.

To that end,  they developed a meta-algorithm that robustifies a given online optimizer by projecting its suggested action onto a robustified action set computed from a given expert method and historical observations.
They provide a worst-case analysis for the cost of the resulting algorithm, based on which, they further establish a robustness guarantee and consistency result in the language of $CR$-competitive (see Definition 4) once a condition is satisfied.
In the appendix, they explore the empirical performance of the proposed algorithm using a case study of battery management and validate its robustness.

**Strengths:**

1. The studied setting is general enough and can incorporate popular neural networks as its part.
2. The worst-case theoretical analysis seems new and the proof seems correct.
3. Though put in the appendix, the experiments are well conducted with detailed descriptions of the experiment setup and chosen parameters, and a careful analysis of the experiment results. I appreciate the experiments a lot.
4. The paper is well-written mostly and the analysis is always to the point.


**Weaknesses:**

However, there are some parts not clear enough.

1. The term ``multi-step nonlinear memory’’ appears abruptly in line 38. I think several additional explanations would help readers to digest this concept better for the first time.

2. In line 200, the author didn’t explain the meaning of $\text{cost}(x_{1:T})$.

3. Line 353, it is unclear to me what the notation $\text{Rob}_{\lambda} \left(\tilde{x}_{1: T} \right)$ means.


**Questions:**

1. Intuitively, once establishing the worst-case bound uniformly in Theorem 4.1, one could easily get an average version (because the average case is always no worse than the worst case). Hence, it is unclear to me why the author bothered to introduce another result to bound the average cost in Theorem 4.2.

2. In Appendix D, the author spent much space explaining how to compute $\nabla_W \mathrm{cost}(x_{1:T})$. The reason is that the operator of projecting ML predictions into a robust set cannot be easily differentiated as typical neural network layers. So why bother to use this projection method? Why not use a primal-dual method that provides a projection-free alternative?

**Limitations:**

See the weakness part and questions.

---

> ### Author Rebuttal · Authors · 2023-08-02
>
> We appreciate your efforts and time in reviewing our paper.
>
> `Average cost bound:` The reviewer is correct that an average cost bound can be established based on the worst cost bound. We provide Theorem 4.2 to confirm that it’s beneficial to train the ML model in a robustification-aware manner according to Eqn. (6). Additionally, it implies that the robustification step is more valuable in terms of bounding the average cost when the ML model is not well trained (e.g., due to inadequate training data). More explanations of the average cost analysis are available in Line 368-387.
>
> `Differentiable robustification:` While adding a regularizer is an alternative to differentiating the constrained optimization (i.e., the robustification step in Line 5 of Algorithm 1), we need to properly tune the weight (i.e., Lagrangian multiplier or dual variable) to meet the constraint. Thus, we directly consider differentiating the robustfication step to be consistent with our algorithm design. The robustification step can be viewed as an implicit layer, which is also differentiatiable. Specifically, to facilitate standard training using back-propagation, we have derived the gradients in Appendix D. Note also that differentiating the implicit layer is also commonly considered in the literature (e.g., the reference at the bottom of this response).
>
> `Multi-step nonlinear memory:` It means the current action appears in the future switching costs for multiple steps, and hence penalizes temporal changes in actions over multiple steps. For example, given the position of a robot as the action for motion planning, a two-step memory captures the acceleration (see details in Ref [2] at the bottom of this response). We’ll explain this in the revised paper.
>
> `Notation:` $cost(x_{1:T})$ means the total cost given a sequence of actions $x_{1:T}$. The notation in Line 353 means the training loss is defined directly in terms of the robustified actions, rather than pre-robustification ML predictions $\tilde{x}_{1:T}$. Hence, we refer to training using this loss as robustification-aware training (Line 348).
>
>
>
> **Reference:**
>
> [1] Akshay Agrawal, Brandon Amos, Shane Barratt, Stephen Boyd, Steven Diamond, and J. Zico 560 Kolter. Differentiable convex optimization layers. NeurIPS 2019.
>
> [2] Shi, Guanya, Yiheng Lin, Soon-Jo Chung, Yisong Yue, and Adam Wierman. Online optimization with memory and competitive control. NeurIPS 2020.

---

> > ### Comment · Reviewer_atug · 2023-08-18
> >
> > I have read the rebuttal. And I keep my score currently.

---

> > > ### Author Response · Authors · 2023-08-18
> > >
> > > We thank the reviewer for reading our rebuttal and hope that the reviewer's concerns have been satisfactorily addressed. We'd also be glad to engage in more discussions to address any remaining concerns that the reviewer may have.

---

### Official Review · Reviewer_6vwR · 2023-06-30

**Soundness:** 2 fair
**Presentation:** 3 good
**Contribution:** 3 good
**Rating:** 6
**Confidence:** 2

**Summary:**

This paper considers the SOCO problem with multi-step nonlinear switching costs and feedback delay. The authors propose the RCL algorithm, which projects the ML predictions on to a robust action set determined by the expert’s predictions. They show that RCL maintain the $(1+\lambda)$-competitiveness against the expert while exploiting the potential of ML predictions. Moreover, they identify a sufficient condition under which the RCL achieves finite robustness and 1-consistency simultaneously. They also give an upper bound of the average cost of RCL, which reveals the advantage of training the ML model in a robustification-aware manner. Finally, they validate the theoretical results using a case study of battery management.

**Strengths:**

- Challenging setup. This paper considers SOCO with hitting cost feedback delay and multi-step nonlinear memory in the switching cost, which is very hard to deal with.
- New algorithm design and new proof approach. The intuition behind their algorithm is clearly explained in line 232-240. Theorem 4.1 is proven by considering a new reservation cost, which decouples the dependency of the online action on the history.
- Good theorems with adequate explanation. The paragraphs following Theorem 4.1, Corollary 4.1.1 and Theorem 4.2 give comprehensive analysis of the results.


**Weaknesses:**

- Assume known switching cost and smoothness constant. Convexity and smoothness assumptions on the hitting costs and switching costs.
- The robustification-aware ML model is hard to be trained using standard back-propagation.


**Questions:**

- Few lines explaining why the cost difference and the switching cost difference can be upper bounded by $H$ and $G$ respectively would help understanding.
- How to efficiently compute the robustified action set? If it cannot be efficiently computed, how you learn that set in your experiment?


**Limitations:**

The authors have already mentioned in the paper (line 408-417).

---

> ### Author Rebuttal · Authors · 2023-08-02
>
> We appreciate your efforts and time in reviewing our paper.
>
> `Explanation of costs difference bounds:` The smoothness of the cost functions implies that, given a bounded difference in two actions, their cost difference is also bounded (formally stated shown in Lemma C.1 in the appendix). Thus, when RCL chooses a different action than the expert, the future cost difference can be bounded in terms of the action difference shown in $H$ and $G$ in Section 4.1.
>
> `Computing the robust action set:` The constrained problem in (1) for the robust action set is convex, and hence it can be efficiently solved using standard convex solvers without a significant computational burden at runtime. For example, in our experiment, each testing process (including 1400+ inferences) takes less than a second on a laptop (Line 586 in the appendix).
>
> `Assumptions:` Along with known switching costs and constants, the convexity and smoothness assumptions on the costs are standard in the SOCO literature. These assumptions are needed for theoretical analysis and can be good approximations of realistic scenarios in practice.
>
> `Training using back-propagation:` The robustification step in Line 5 of Algorithm 1 can be viewed as an implicit layer, which can also be differentiated. Specifically, to facilitate standard training using back-propagation, we have derived the gradients in Appendix D. Note also that differentiating the implicit layer is also commonly considered in the literature (e.g., the reference at the bottom of this response).
>
> **Reference:**
>
> Akshay Agrawal, Brandon Amos, Shane Barratt, Stephen Boyd, Steven Diamond, and J. Zico 560 Kolter. Differentiable convex optimization layers. NeurIPS 2019.

---

> > ### Comment · Reviewer_6vwR · 2023-08-12
> > **Thanks for the rebuttal**
> >
> > I have read the rebuttal and the other reviews. And I will keep my score.

---

> > > ### Author Response · Authors · 2023-08-13
> > >
> > > We thank the reviewer for reading our rebuttal and hope that the reviewer's concerns have been satisfactorily addressed. We'd also be glad to engage in more discussions to address any remaining concerns that the reviewer may have.

---

### Official Review · Reviewer_WzoZ · 2023-07-02

**Soundness:** 3 good
**Presentation:** 2 fair
**Contribution:** 2 fair
**Rating:** 5
**Confidence:** 2

**Summary:**

This paper considers the smoothed online convex optimization problem, in which, in addition to the standard hitting cost incurred per step, the algorithm also incurs a switching cost which relates to changes in its chosen action. The paper considers switching costs which are multi-step (relate to the previous p time steps).

The paper presents an ML-augmented algorithm for the problem, which given a possibly-inaccurate predictor and a robust expert, is able to utilize the predictions from the predictor to the degree to which they are accurate, while not exceeding the cost of the expert times some small factor. This is done by taking the predictions and projecting them onto a space of possible actions dictated by the expert.

The paper also discusses how to adjust the training of an ML model to take into account the fact that the model's predictions are then projected onto the actions allowed by the expert.

**Strengths:**

The model, and the concept underlying the algorithm, are natural.
I liked the figure in Appendix A, perhaps it should be part of the main body of the paper.

**Weaknesses:**

In my opinion, the writing of the paper makes it hard to gain an intuitive understanding of the contributions of the paper within the scope of reasonable reviewing. For example, consider Theorem 4.1; excluding the robustness term, it is very hard to understand the significance of this theorem. Perhaps this bound seems simple and intuitive to experts more familiar with this problem.

**Questions:**

none

**Limitations:**

yes

---

> ### Author Rebuttal · Authors · 2023-08-02
>
> We appreciate your efforts and time in reviewing our paper.
>
> `Significance:` We’ll revise the paper to improve its presentation. Our contribution is a novel ML-augmented algorithm that exploits ML predictions to improve the average performance while bounding the worst-case performance in a general SOCO setting with hitting cost feedback delay and multi-step non-linear switching costs. The feedback delays introduce additional uncertainties, and multi-step memory in the switching cost means the current action can affect future costs in multiple steps. Both make it substantially challenging to construct a robust action set. In fact, even without ML predictions, addressing the hitting cost feedback delay and multi-step nonlinear memory is already challenging and an independent study [26].
>
> Our algorithm design and analysis (Theorem 4.1) are substantially new and differ from those used in simple SOCO settings. More specifically, the proof of Theorem 4.1 relies on a novel and crucial techniques that removes the dependency of $x_t$ on the history in the online decision process.
>
> The second term in the bound in Theorem 4.1 shows how well RCL follows the ML predictions given $\lambda>0$. Specifically, when $\lambda$ increases, RCL will be closer to ML (see the definition of $\Delta(\lambda)$ in Line 252). Moreover, it shows that RCL stays closer to the better-performing expert for guaranteed competitiveness when the expert’s cost is lower, and vice versa.
>
> **To summarize, our work is the first learning-augmented algorithm for the challenging SOCO setting with feedback delay and multi-step switching costs. It makes a novel and significant contribution to the growing SOCO literature.**

---

> > ### Comment · Reviewer_WzoZ · 2023-08-15
> >
> > Thank you for your answer.

---

> > > ### Author Response · Authors · 2023-08-15
> > >
> > > We thank the reviewer for reading our rebuttal and hope that the significance and novelty of our work have been clarified. We'd also be glad to engage in more discussions to address any remaining concerns that the reviewer may have.

---

### Official Review · Reviewer_rcGc · 2023-07-23

**Soundness:** 3 good
**Presentation:** 3 good
**Contribution:** 2 fair
**Rating:** 5
**Confidence:** 4

**Summary:**

This paper studies the  Smoothed Online Convex Optimization with  multi-step nonlinear switching costs and feedback delay. In this setting, they propose Robustness-Constrained Learning (RCL), which  combines  existing online algorithm with a novel reservation cost to robustify untrusted ML predictions. Theoretically, they provide both  worst-case and average-case guarantees  for RCL. Finally, they provide some experiments to evaluate RCL.

**Strengths:**

1. The paper is well written and organized. The paper thoroughly explains the challenges of incorporating ML predictions in their setting and emphasizes the importance of developing novel algorithmic techniques to overcome these difficulties.

2. The theoretical results are sound and look rigorous. The worst-case bound of RCL is novel.


**Weaknesses:**

1. The weakness of the paper lies in the design of reservation costs, which may seem somewhat technical for analysis purposes. Additionally, the discussion on the optimization complexity for solving the constrained convex problem is missing, leaving a gap in understanding the practical implications of the proposed approach.

2. The paper's experimental results should be included in the main part to enhance confidence in the efficiency of RCL. Presenting empirical comparisons of RCL with other methods would demonstrate its advantages and substantiate the claims made in the paper. This would provide valuable evidence of the proposed algorithm's performance and practical benefits.


**Questions:**

See Weakness

**Limitations:**

There are no limitations with regards to negative societal impacts.

---

> ### Author Rebuttal · Authors · 2023-08-02
>
> We appreciate your efforts and time in reviewing our paper.
>
> `Reservation cost and complexity:` If we choose an action different from the expert without a reservation cost, it’s possible that the competitiveness against the expert is violated (an example is provided in Lines 210-217). Thus, for guaranteed competitiveness, our novel reservation costs are designed to hedge against any possible uncertainties due to hitting cost feedback delays and multi-step non-linear memory in the switching costs. It bounds the maximum future cost difference in terms of the difference between our actual action and the expert's action. We’ll revise the presentation for the reservation cost to improve its readability.
>
> The constrained problem in (1) for computing the robust action set is convex, and hence it can be efficiently solved using standard convex solvers without a significant computational burden at runtime. For example, in our experiment, each testing process (including 1400+ inferences) takes less than a second on a laptop (Line 586 in the appendix).
>
> `Simulation results:` Our experimental results for EV charging station management can be found in Appendix B. Our results highlight the advantage of RCL in terms of robustness guarantees compared to pure ML models, as well as the empirical benefit of training a robustification-aware ML model in terms of the average cost. Per the reviewer’s suggestion, we’ll include some of the key results in the main body of the paper for better understanding.

---

> > ### Comment · Reviewer_rcGc · 2023-08-15
> >
> > Thanks for the rebuttal. After reading the rebuttal and the other reviews,  I decided to keep my original score.

---

> > > ### Author Response · Authors · 2023-08-15
> > >
> > > We thank the reviewer for reading our rebuttal and hope that the reviewer's concerns have been satisfactorily addressed. We'd also be glad to engage in more discussions to address any remaining concerns that the reviewer may have.

---

### Official Review · Reviewer_axLK · 2023-07-24

**Soundness:** 2 fair
**Presentation:** 3 good
**Contribution:** 2 fair
**Rating:** 5
**Confidence:** 3

**Summary:**

This paper studies a problem, namely, Smoothed Online Convex Optimization (SOCO) with feedback delay and multi-step nonlinear memory, proposing a machine learning augmented online algorithm (RCL) with certain theoretical guarantees. This algorithm considers to implicitly hedge the decisions made by online and offline predictors, using a new constraint action set onto which the algorithm projects its decisions. This constraint set requires the decisions within it to achieve a trade-off between the hitting cost and the switching cost. By this design, this work achieves the first worst-case type cost bound for machine learning augmented algorithm under the given setting, and proposes a sufficient condition for the proposed algorithm to attain finite robustness and 1-consistency simultaneously. Moreover, the paper demonstrates that the algorithm RCL benefits from robustification-aware training and offers an average cost guarantee. Comprehensive experiments are executed to support these results.

**Strengths:**

The writing of this paper is good with detailed explanations of the achieved results and the motivation is well substantiated by the experiments. The proposed algorithm is clear and easy to understand. The inherent nature of delayed feedback in the considered problem might pose an obstacle for algorithm design and analysis, and this paper proposes a solution for it.

**Weaknesses:**

I harbor some reservations regarding the applicability and scalability of the theoretical findings presented in this paper. Please refer to the "Questions" section below for further details.

**Questions:**

1.	This paper claims that it provides a sufficient condition to achieve the finite robustness and 1-consistency simultaneously (line 63). But Corollary 4.1.1 suggests that this condition might be applicable for the proposed specific algorithm only. Moreover, this condition requires a per round lower bound for the switching cost of the expert algorithm. Does this condition hold for a broad class of expert algorithms? Could the sufficient condition proposed in Corollary 4.1.1 be suitable for a more general ML-augmented algorithm to realize the best-of-both-world guarantee?
2.	I am confused about the term "worst-case" used in the discussion about Theorem 4.1. Based on my understanding, the worst-case bound should account for the upper bound of the minimax ratio. It would be beneficial if the authors could discuss about the relationship between the minimax bound and the bound in Theorem 4.1.
3.	There are concerns regarding the existence and convexity of $\mathcal{X}_t$. Does the solution of Eq. (1) below line 223 always exist? It appears from Corollary 4.1.1 that the action set $\mathcal{X}$ is finite. If that is the case, then the "robustified" action set defined at line 223 may not exhibit convexity due to its discrete nature. Can the authors answer the above two questions?
4.	Would the authors be able to provide further discussion concerning the technical contributions made in addressing the issues related to feedback delay, especially in comparison with previous works?

**Limitations:**

Not much.

---

> ### Author Rebuttal · Authors · 2023-08-06
>
> We appreciate your efforts and time in reviewing our paper.
>
> `Sufficient condition:` The prior study [1] proves an "impossibility" result: 1-consistency and finite robustness are not achievable simultaneously without further assumptions due to the fundamental hardness of the SOCO problem (even without delay feedback or multi-step memory). Our result in Corollary 4.1.1 shows a sufficient condition for our algorithm (RCL) to achieve 1-consistency and finite robustness simultaneously. There might exist other conditions under which alternative algorithms proposed in the future may also achieve 1-consistency and finite robustness simultaneously, but our work provides the first condition in the field and advances the result in [1].
>
> Our sufficient condition is not unrealistic in practice (Line 329-332). In fact, the condition is violated by an expert (not necessarily the best-known competitive expert iROBD) only in dummy cases (i.e., its action that results in zero switching cost also has zero hitting cost). Even when the condition is not satisfied, our main result in Theorem 4.1 still bounds the cost of RCL and is consistent with the "impossibility" result in [1].
>
> `Minimax and Theorem 4.1:` In online optimization, the competitive ratio is typically defined as the maximum cost ratio of one algorithm to another (Definition 2). Thus, the "worst case" means the case that results in the maximum cost ratio. "Minimax" arises when minimizing the competitive ratio (i.e., minimizing the maximum cost ratio). The "worst case" in Line 256 means that our bound in Theorem 4.1 applies for any possible problem instances, including those worst-case instances that have the maximum cost ratios.
>
> `Existence and convexity of` $\mathcal{X}_t$: An improperly-constructed robust action set can result in empty solution sets (example in Lines 207-217). Thus, we make contributions by designing a reservation cost that ensures there always exists a non-empty solution set regardless of future uncertainties. This is also stated in Lines 241-242 and proved in Lemma C.4 in the appendix.
>
> As in the standard SOCO literature (e.g., [1]), the action set $\mathcal{X}$ is assumed to be a subset of $R^n$ (Line 116), and its size $|\mathcal{X}|$ can be measured by the maximum norm distance between two points in $\mathcal{X}$ (e.g., [3,4]). Thus, along with convex costs, the set $\mathcal{X}_t$ is also convex. We'll clarify this point in our revision.
>
> `Feedback delay`: The feedback delays introduce significant uncertainties for robustness guarantees, as we cannot immediately evaluate the costs of our algorithm and the expert. Thus, for robustness guarantees, we have to consider additional risks due to not following the expert's action and bound the maximum cost difference in Eq. (2) by exploiting the cost structures. The prior study [2] also considers multi-step feedback delays, but it has a more restrictive model where all the feedbacks have the same delay (whereas we allow different delays of up to $q$ steps in Definition 1 in Line 154-157). Most importantly, [2] is a purely expert algorithm and does not consider an ML prediction. To our knowledge, our work is the first to consider learning-augmented algorithms for the challenging setting SOCO with feedback delays.
>
>
> **References:**
>
> [1] Daan Rutten, Nicolas Christianson, Debankur Mukherjee, and Adam Wierman. 2023. Smoothed Online Optimization with Unreliable Predictions. Proc. ACM Meas. Anal. Comput. Syst. 7, 1, Article 12 (March 2023).
>
> [2] Weici Pan, Guanya Shi, Yiheng Lin, and Adam Wierman. 2022. Online Optimization with Feedback Delay and Nonlinear Switching Cost. Proc. ACM Meas. Anal. Comput. Syst. 6, 1, Article 17 (March 2022).
>
> [3] Marek Bukáček, Pavel Hrabák, Milan Krbálek. 2018. Microscopic Travel Time Analysis of Bottleneck Experiments. Transportmetrica A Transport Science.
>
> [4] https://engineering.purdue.edu/ChanGroup/ECE302/files/Slide_4_01.pdf

---

> > ### Author Response · Authors · 2023-08-18
> >
> > We thank you for your valuable time and effort in reviewing our paper. We hope that our responses have satisfactorily addressed your concerns and provided a better clarification of our contribution. We would genuinely appreciate your responses or comments should there be any remaining concerns, and are more than happy to address them. Once again, thank you for your valuable time in reviewing our paper.

---

> > > ### Comment · Reviewer_axLK · 2023-08-19
> > >
> > > I thank the authors for answering my questions. And I decide to raise my score to 5.

---

> > > > ### Author Response · Authors · 2023-08-19
> > > >
> > > > We thank the reviewer for reading our rebuttal and are glad that the reviewer's concerns have been addressed. We'd very much like to continue discussions with the reviewer if the reviewer has any remaining concerns.

---

### Official Review · Reviewer_6WgB · 2023-07-25

**Soundness:** 3 good
**Presentation:** 3 good
**Contribution:** 3 good
**Rating:** 5
**Confidence:** 4

**Summary:**

This paper studies Smoothed Online Convex Optimization with multi-step nonlinear switching costs and feedback delay. They propose an ML-augmented online algorithm named RCL that combines ML predictions with an expert online algorithm. The authors show that RCL can guarantee $(1 + \lambda)$ competitive ratio against any expert while improving the average-case performance. They show the effectiveness of RCL using battery management as a case study.

**Strengths:**

1. Learning augmented algorithms is an important paradigm for designing more practical algorithms. Incorporating ML-augmented algorithms with multi-step nonlinear switching costs and feedback delay is a new and challenging problem.
2. The algorithm is simple and intuitive to understand. The authors provide a detailed analysis of the algorithm and the proof is rigorous.
3. This paper is well-written and easy to follow. The authors provide a detailed literature review and a clear introduction to the problem setting. The authors also provide a detailed discussion of the experimental results.

**Weaknesses:**

1. The algorithmic contributions of the paper are a bit weak. The idea of constructing reservation costs to hedge against any possible uncertainties and then solving a constrained convex problem to project the ML prediction into a robust action set is quite standard in ML-augmented algorithms.
2. There lacks a lower bound on the trade-offs between robustness and average performance in ML-augmented algorithms.

**Questions:**

Is it possible for the reservation cost design to be overly conservative, resulting in the constrained optimization problem yielding an empty solution set?

---

> ### Author Rebuttal · Authors · 2023-08-04
>
> We appreciate your efforts and time in reviewing our paper.
>
> `Algorithm design and novelty:` It’s essential to construct a robust action set to hedge against future uncertainties for guaranteed robustness in learning-augmented online algorithms. Nonetheless, how to achieve so is very challenging. In our problem, the feedback delays introduce additional uncertainties, and multi-step memory in the switching cost means the current action can affect future costs in multiple steps. Both make it substantially challenging to construct a robust action set. In fact, even without ML predictions, addressing the hitting cost feedback delay and multi-step nonlinear memory is already challenging and an independent study. For the first time, our carefully-desgined reservation cost guarantees competitiveness of unreliable ML predictions in this challenging SOCO setting. To our best knowledge, this is a novel and significant contribution to the SOCO literature.
>
> `Empty solution set:` An improperly-constructed robust action set can result in empty solution sets (example in Lines 207-217). This also further reinforces our previous point "how to achieve so (i.e., constructing a robust action set) is very challenging". Therefore, we make contributions by designing a reservation cost that ensures there always exists a **non-empty** solution set regardless of future uncertainties. This is also stated in Lines 241-242 and proved in Lemma C.4 in the appendix.
>
> `Lower bound:` The performance analysis for learning-augmented online algorithms involves a tradeoff between following the ML predictions for average performance improvement and following the expert for robustness (as governed by $\lambda>0$ in our paper). To our best knowledge, the optimal (or "lower bound") tradeoff still remains an open challenge in learning-augmented algorithms, except for a small number of simple online problems. Our learning-augmented algorithm is the first and makes contributions to the general SOCO setting with delayed feedback and multi-step memory. Our result in Theorem 4.1 bounds the performance of RCL, and shows the clear insight that the robustness parameter $\lambda>0$ governs the tradeoff between following ML predictions and the expert's actions. While it's important to characterize the Pareto-optimal tradeoff curve, we leave it as future work that the learning-augmented algorithm community can explore (Line 415).

---

> > ### Comment · Reviewer_6WgB · 2023-08-14
> >
> > After reviewing the rebuttal and considering the other reviewers' feedback, I have decided to maintain my original score.

---

> > > ### Author Response · Authors · 2023-08-14
> > >
> > > We thank the reviewer for reading our rebuttal and hope that the reviewer's concerns (e.g., non-empty solution sets guaranteed by our novel algorithm design) have been satisfactorily addressed. We welcome any additional comments the reviewer may have and would be glad to address them accordingly.

---

### Author Response · Authors · 2023-08-21

Dear Reviewers,

We thank you for your valuable time in reviewing our paper and responding to our rebuttals. Your comments were greatly appreciated. We are very glad that our responses have provided a better clarification of our important contribution to the growing field of learning-augmented smoothed online convex optimization (SOCO).

Both **feedback delay** and **multi-step memory** introduce significant *algorithmic challenges* and require an independent study even without considering ML predictions. For the challenging SOCO setting with feedback delay and multi-step memory, our learning-augmented algorithm utilizes a *novel design* of efficiently-computable online action sets to provably guarantee the worst-case competitiveness, while being able to explore the potential of ML predictions to improve the average performance. In addition to the theoretical analysis, we also derive gradients of the introduced robustification step to facilitate ML model training, and our experimental results based on online electric vehicle charging demonstrate the empirical advantage of our algorithm compared to existing solutions.

As the author-reviewer discussion phase is approaching to the end, we are more than happy to address any remaining concerns that you may have. Once again, we thank you for your valuable comments and engaging in discussions with us.

Authors of Paper 3837

---

### Decision · Program_Chairs · 2023-09-21

**Decision:**

Accept (poster)

**Comment:**

In this paper, the authors investigate a variant of smoothed online convex optimization (SOCO) which includes multi-step nonlinear switching costs and feedback delay. To this end, they propose a machine learning (ML) augmented online algorithm, namely RCL, which utilizes an expert algorithm to robustify untrusted ML prediction. They prove that RCL is able to guarantee (1+$\lambda$)-competitiveness against any expert while improving the average-case performance.

This paper is easy to follow, and the algorithm is well-motivated. RCL is the first ML-augmented algorithm with a provable robustness guarantee under the setting of this paper. This constitutes a significant contribution to the field of SOCO.